# Generation and optimization of off-the-shelf immunotherapeutics targeting TCR-Vβ2+ T cell malignancy

Jingjing Ren [1,4] ✉, Xiaofeng Liao[1,4] ✉, Julia M. Lewis[1], Jungsoo Chang [1], Rihao Qu[2], Kacie R. Carlson[1], Francine Foss [3] & Michael Girardi [1] ✉

Current treatments for T cell malignancies encounter issues of disease relapse and off-target toxicity. Using T cell receptor (TCR)Vβ2 as a model, here we demonstrate the rapid generation of an off-the-shelf allogeneic chimeric antigen receptor (CAR)-T platform targeting the clone-specific TCR Vβ chain for malignant T cell killing while limiting normal cell destruction. Healthy donor T cells undergo CRISPR-induced *TRAC*, *B2M* and *CIITA* knockout to eliminate T cell-dependent graft-versus-host and host-versus-graft reactivity. Second generation 4-1BB/CD3zeta CAR containing high affinity humanized anti-Vβ scFv is expressed efficiently on donor T cells via both lentivirus and adeno-associated virus transduction with limited detectable pre-existing immunoreactivity. Our optimized CAR-T cells demonstrate specific and per-sistent killing of Vβ2+ Jurkat cells and Vβ2+ patient derived malignant T cells, in vitro and in vivo, without affecting normal T cells. In parallel, we generate humanized anti-Vβ2 antibody with enhanced antibody-dependent cellular cytotoxicity (ADCC) by Fc-engineering for NK cell ADCC therapy.

T cell malignancies are a clinically and biologically heterogeneous group of disorders, together comprising ~10–20% of non-Hodgkin's lymphomas and ~20% of acute leukemias. A challenge in the develop-ment of effective and safe immunotherapies for T cell malignancies and T cell disorders is the considerable overlap in marker expression present on both the target clonal and off-target T-cells, with no single antigen clearly able to identify clonal (malignant) cells[1]. Chimeric antigen receptor (CAR) T-cell therapy and therapeutic antibody tar-geting CD19 have shown efficacy in the treatment of B cell leukemias and lymphomas[2,3] and B cell-driven autoimmunity (e.g. pemphigus vulgaris, systemic lupus erythematosus)[4,5], but commonly also results in substantial depletion of a patient's normal B cell population. While the effects of B cell depletion may be clinically managed, e.g. with lifelong intravenous immunoglobulin infusions, similar levels of T cell depletion may render patients at substantial risk for life-threatening infections[6]. Although potential CAR-T therapies targeting different

surface markers of malignant T cells have been developed and several have proceeded into clinical trials[1,7], this approach must contend with potential pitfalls of immune suppression by T cell-depletion (by affecting normal T cells), CAR-T self-killing (if the target-antigen is expressed on the CAR-T), and CAR-T transfection of malignant cells (if the patient is the source of the T cells for CAR transduction). Several antibody therapies for T malignancy have also been approved for clinical use but patients frequently develop resistance and relapse, and high rates of opportunistic infection are reported[8]. Therefore, a need exists for innovative strategies in the treatment of T cell malignancies where therapeutic agents exhibit high specificity and efficacy in the elimination of disease-mediating T cells.

Across the spectrum of T cell leukemias and lymphomas—e.g. cutaneous T cell lymphoma (CTCL), peripheral T cell lymphoma (PTCL), anaplastic large cell lymphoma (ALCL), adult T cell leukemia/lymphoma (ATLL), T-cell large granular T cell lymphocytic leukemia

[1]Department of Dermatology, Yale School of Medicine, New Haven, CT, USA. [2]The Computational Biology and Bioinformatics Program, Yale School of Medicine, New Haven, CT, USA. [3]Department of Internal Medicine, Section of Medical Oncology, Yale School of Medicine, New Haven, CT, USA. [4]These authors contributed equally: Jingjing Ren, Xiaofeng Liao. ✉e-mail: jingjing.ren@yale.edu; xiaofeng.liao@yale.edu; michael.girardi@yale.edu

(T-LGLL), T-cell acute lymphocytic leukemia (T-ALL), T cell chronic lymphocytic leukemia (T-CLL)—the malignant T cells invariably express a single (or rarely oligo) dominant T cell receptor(s) that allows for the potential therapeutic targeting via Vβ-family specific CAR-T and antibody therapies[9–13]. Moreover, such a patient-matched Vβ-specific immunotherapy would be predicted to have limited effects on normal T cells since each of the Vβ genes is expressed by only a small fraction (~1–10%) of the overall T cell population. Although there are 45 functional Vβ gene fragments in human genome[14], the currently commercially available set of 24 Beckman Coulter anti-human Vβ monoclonal antibodies (mAb) identifies 70–80% of total T cells[15–17]. The development of a uniform product pipeline together with a personalized anti-Vβ application adjustment will enable therapeutic strategies based on the anti-Vβ mAb set to maximize their potential and flexibility to treat T cell malignancy, and potentially pathogenic T cell-dependent autoimmune diseases[18,19].

Flow cytometry studies using the anti-Vβ mAb set have shown that Vβ2 is among the most frequently used Vβ segments, at 9% of T cells of healthy individuals[15]. Moreover, multiple studies have shown that clonal Vβ2 usage is most common among established human cancer T cell lines and mature T cell lymphoma patients[12,16]. Another study focusing on T-LGLL also found the incidence of clonal Vβ2 usage among patients to be the highest, at 23.5%[17], and we have similarly observed that Vβ2 is the most common Vβ among CTCL patients at Yale (below).

In this work, using TCR-Vβ2 as a prototype therapeutic target towards the potential for the development of a fuller set of anti-Vβ therapeutics that might be selectively matched to patients with T cell malignancy, we demonstrate herein the feasibility of a strategy for the development of off-the-shelf humanized allogeneic anti-Vβ2 CAR-T cells, and show the feasibility of allogeneic NK cell-dependent humanized ADCC-enhancing antibodies, for the potential specific treatment of Vβ2+ T cell malignancies.

## Results

### Generation of CAR-T cells targeting TCR Vβ2+ on malignant T cells

Sampling CTCL patients with peripheral blood involvement, we observed that malignant cells expressed a clone-defining TCR using a single Vβ-family, the most frequent of which was Vβ2 (TRBV20-1 detected by anti-Vβ2 antibody clone MPB2D5) (Fig. 1a). Through paired single cell mRNA/TCR sequencing, we previously demonstrated a single dominant TCR clone of CTCL cells in each of the 11 CTCL patients investigated[13], among which the distribution of the top ten TRBV (T cell receptor beta variable region) subtypes in a Vβ2+ patient is shown in Fig. 1b. The single dominant Vβ2 clonality of malignant T cells was further validated by flow cytometry in the same Vβ2+ CTCL patient (Fig. 1b, Supplementary Fig. 1a). Therefore, targeting dominant TRBVs by corresponding Vβ antibodies and/or their therapeutic derivatives, such as CAR-T cells, may be a specific way to treat T malignancy without substantially affecting normal T cells. We initially utilized an available murine Vβ2 antibody for the development of our anti-Vβ CAR-T cell platform due to the frequency of Vβ2+ CTCL cases (Fig. 1a). The antibody amino acid sequence was obtained by mass spectrometry, followed by the incorporation of VH and VL sequences into a second-generation CAR construct (mCAR-Vβ2)[20] (Supplementary Fig. 1b), with a two cysteine replacement at Kabat position VH44 and VL100 to increase the scFV stability[21,22]. To generate a more accessible target cell model, we overexpressed human TCR beta chain containing TRBV20-1 (Supplementary Fig. 1c) in Jurkat cells after knockout of the original TCR beta chain (Jurkat-TRBV20-1). Surface CAR expression on healthy donor CD8 T cells was positively correlated to GFP reporter expression (Supplementary Fig. 1d), suggesting that intracellular GFP expression level accurately reflected T cell surface expression of the CAR protein. Similarly, intracellular mCherry acted as a surface TRBV20-1 expression reporter on Jurkat-TRBV20-1 cells

(Supplementary Fig. 1e). Relative to control CAR-CD19 T cells, mCAR-Vβ2 T cells generated from the same healthy donor induced significantly higher killing of Jurkat-TRBV20-1 cells when co-cultured overnight at an effector to target (E:T) ratio of 1:1 (Supplementary Fig. 1f), suggesting an efficient binding affinity of mCAR-Vβ2 to TRBV20-1 through the recombinant scFv domain. However, CAR-CD19 T cells from the healthy donor also showed some Jurkat-TRBV20-1 cell killing (Supplementary Fig. 1f), consistent with the potential for alloreactivity of donor CAR-T cells. Thus, we also developed an autologous mCAR-Vβ2 T cell model by using purified CD8+ T cells from a Vβ2+ CTCL patient as the source of mCAR-Vβ2 T cells to target the patient's (CD4+) Vβ2+ CTCL cells. Relative to CD8 T cells from healthy donors, however, those from the CTCL patient showed substantially lower lentiviral transduction efficiency and CAR expression (Supplementary Fig. 1g). Furthermore, we found that autologous mCAR-Vβ2 T cells sourced from three different Vβ2+ CTCL patients' T cells, showed no significant killing of Vβ2+ CTCL cells at an E:T of 1:1, while significant but limited killing was evident at an E:T of 5:1 (Supplementary Fig. 1h). These results suggested that autologous CD8 T cells from T-cell malignant patients may not be an effective source for CAR-T cells. This finding is consistent with published data showing that cell-mediated immunity, including CD8 T cell function, is markedly compromised in CTCL patients[23–25] as well as patients with other T cell malignancies[26,27]. Therefore, we explored the feasibility and importance of generating allogeneic CAR-T cells from healthy donors to increase the efficacy of malignant T cell killing.

### Allogeneic *TRAC/B2M/CIITA* triple-knockout mCAR-Vβ2 T cells show specific Vβ2 targeting while minimizing GVH effects

Towards the development of off-the-shelf allogeneic mCAR-Vβ2 T cells from healthy donors, we firstly knocked out the *TRAC* gene[28] of purified CD8 T cells with over 70% efficiency, followed by residual CD3+ cell depletion (Supplementary Fig. 2a). Without the *TRAC* knockout (KO), mCAR-Vβ2 T cells showed not only on-target killing of Vβ2+ CTCL cells but also off-target killing of control Vβ13.2+ CTCL cells freshly isolated from corresponding patients (Fig. 1c, left). After the elimination of endogenous TCRs on mCAR-Vβ2 T cells, however, this non-specific reactivity was markedly reduced as evidenced by the limited killing of Vβ13.2+ CTCL cells while efficient killing of Vβ2+ CTCL cells was retained (Fig. 1c, right), suggesting that allogeneic mCAR-Vβ2 T cells generated from healthy donors were specific and effective against the appropriate Vβ+ malignant CTCL cells. To increase the potential persistence of allogeneic CAR-Vβ2 cells in patients by reducing HVG reactivity and rejection, *B2M* and *CIITA* genes[29] were subsequently also knocked out after *TRAC* KO (hereafter "triple-KO") in purified pan T cells, with demonstrated down-regulation of HLA-A/B/C and HLA-DR/DP/DQ surface expression (Supplementary Fig. 2b, c). In addition, such a triple-KO did not substantially affect lentiviral transduction, as shown by flow cytometry where CAR-Vβ2+ (GFP+) cells comprised over 70% of both CD4 and CD8 T cells (Supplementary Fig. 2d). Our triple-KO approach also reduces the concern that the generated CAR-Vβ2 T cells will express Vβ2, or become hypofunctional due to activation and elimination of Vβ2+ cells during CAR generation. Thus, we used triple-KO allogeneic CAR-T cells from healthy donors in all the studies presented hereafter.

To further assess specificity, efficacy and potential safety of mCAR-Vβ2 T cells, PBMC from two different Vβ2+ CTCL patients (Fig. 1d) and a healthy donor (Fig. 1e) were co-cultured with mCAR-Vβ2 T cells prepared from a different healthy donor. While Vβ2+ CTCL cells from patients and Vβ2+ normal cells from the healthy donor were significantly killed by the mCAR-Vβ2 T cells, the viability of Vβ2-negative normal T cells and CD3-negative (i.e. non-T) cells was not affected. A full analysis of the Vβ repertoire of T cells from healthy donor 2 further showed that all T cell subpopulations with detectable Vβs other than Vβ2 were relatively spared elimination by mCAR-Vβ2 T

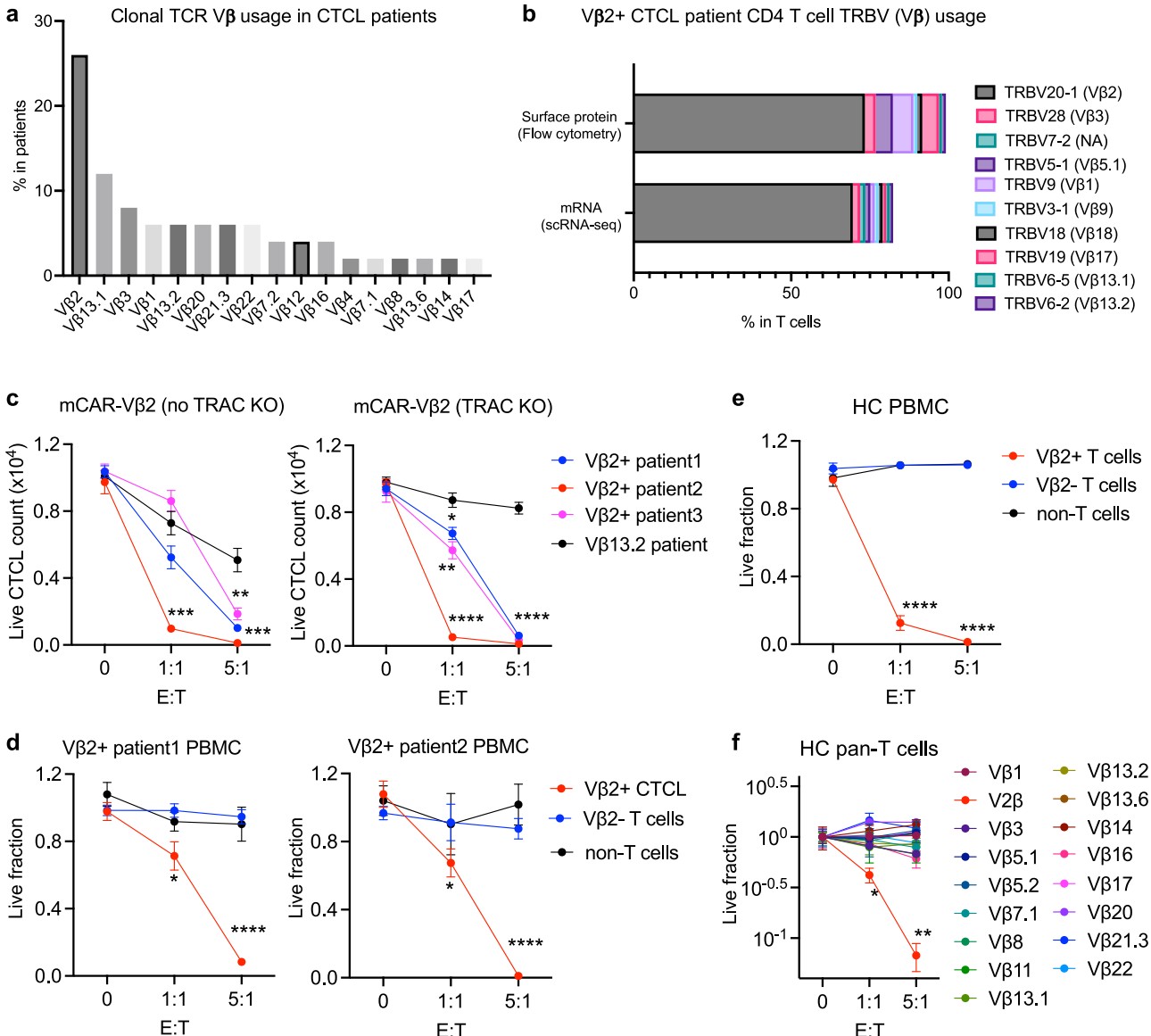

**Fig. 1 | Generation of mCAR-T cells targeting the TCR-Vβ2 chain. a** TRBV usage frequency in CTCL cells from 72 patients seen at the Yale Photopheresis Unit from 2016 to 2022 as determined by anti-Vβ antibody staining (Beckman Coulter IOTest Beta Mark) and flow cytometry. **b** Distribution of TRBV frequency in total CD4 T cells from a Vβ2+ Sézary Syndrome (SS) patient determined by anti-Vβ antibody set staining and flow cytometry (top) or by paired single cell mRNA/TCR sequencing (bottom[13],). **c** Live CTCL cell counts from three Vβ2+ and one Vβ13.2 + CTCL patients after overnight in vitro culture with allogeneic mCAR-Vβ2 T cells with (right) or without (left) knockout (KO) of the endogenous T cell receptor alpha constant (*TRAC*) region at different effector-to-target (E:T) ratios, as determined by flow cytometry. **d, e** Live fraction of Vβ2+ CTCL cells or normal Vβ2+ cells (red), Vβ2- normal T cells (blue), and non-T cells (black) from PBMC of two Vβ2+ CTCL patients (**d**) and healthy control (HC)1 (**e**) at different E:T ratios after overnight co-culture with allogeneic triple KO (*TRAC/B2M/CIITA*) mCAR-Vβ2 T cells, as determined by flow cytometry. **f** Live fraction of each detectable Vβ subtype+ cells from total T cells of HC2 at different E:T ratios after overnight co-culture with allogeneic triple KO mCAR-Vβ2 T cells, as determined by flow cytometry. **c–f** *N* = 3 replicates of each E:T condition. Data are presented as mean values +/− SEM. *p < 0.05, **p < 0.01 and ****p < 0.0001 by two-way ANOVA. All replicates are independent samples. Source data and exact *p*-values are provided as a Source Data file.

cell exposure in vitro (Fig. 1f). Consistent with this specificity was the upregulation of major T cell activation markers−CD137 (4-1BB), CD25 and CD69−on mCAR-Vβ2 T cells after co-culture with Vβ2+ but not Vβ13.2+ CTCL cells (Supplementary Fig. 2e). These data collectively demonstrate the ability of our strategy to generate allogeneic mCAR-Vβ2 T cells to target Vβ2+ malignant T cells efficiently and specifically while minimizing the potential for GVH-related toxicity in vitro.

**Allogeneic mCAR-Vβ2 T cells efficiently and specifically eliminate Vβ2+ malignant T cells in vivo**

To assess the capacity of our mCAR-Vβ2 T cells to target malignant T cells in vivo, we generated a patient-derived xenograft (PDX) model

in NSG mice (Fig. 2a) populated with total CD4 T cells from a Vβ2+ leukemic PTCL patient, or with tumor line Jurkat-TRBV20-1 (Vβ2+) cells. Both malignant cell populated mouse models were treated with mCAR-Vβ2 T cells prepared from purified CD8 T cells of a healthy donor. In the NSG mice repopulated with Jurkat-TRBV20-1 cells, at the endpoint of tissue collection (10 days post i.v. Jurkat-TRBV20-1 transfer), Jurkat-TRBV20-1 cells predominantly accumulated in the bone marrow and showed significantly reduced cell number in the mCAR-Vβ2 T cell treated group relative to the non-treated control (NC) group (Fig. 2b, c), indicating a high on-target killing efficacy of mCAR-Vβ2 T cells in vivo. In addition, mCAR-Vβ2 T cells were persistent in NSG mice at day 7 post intravenous transfer without the need for

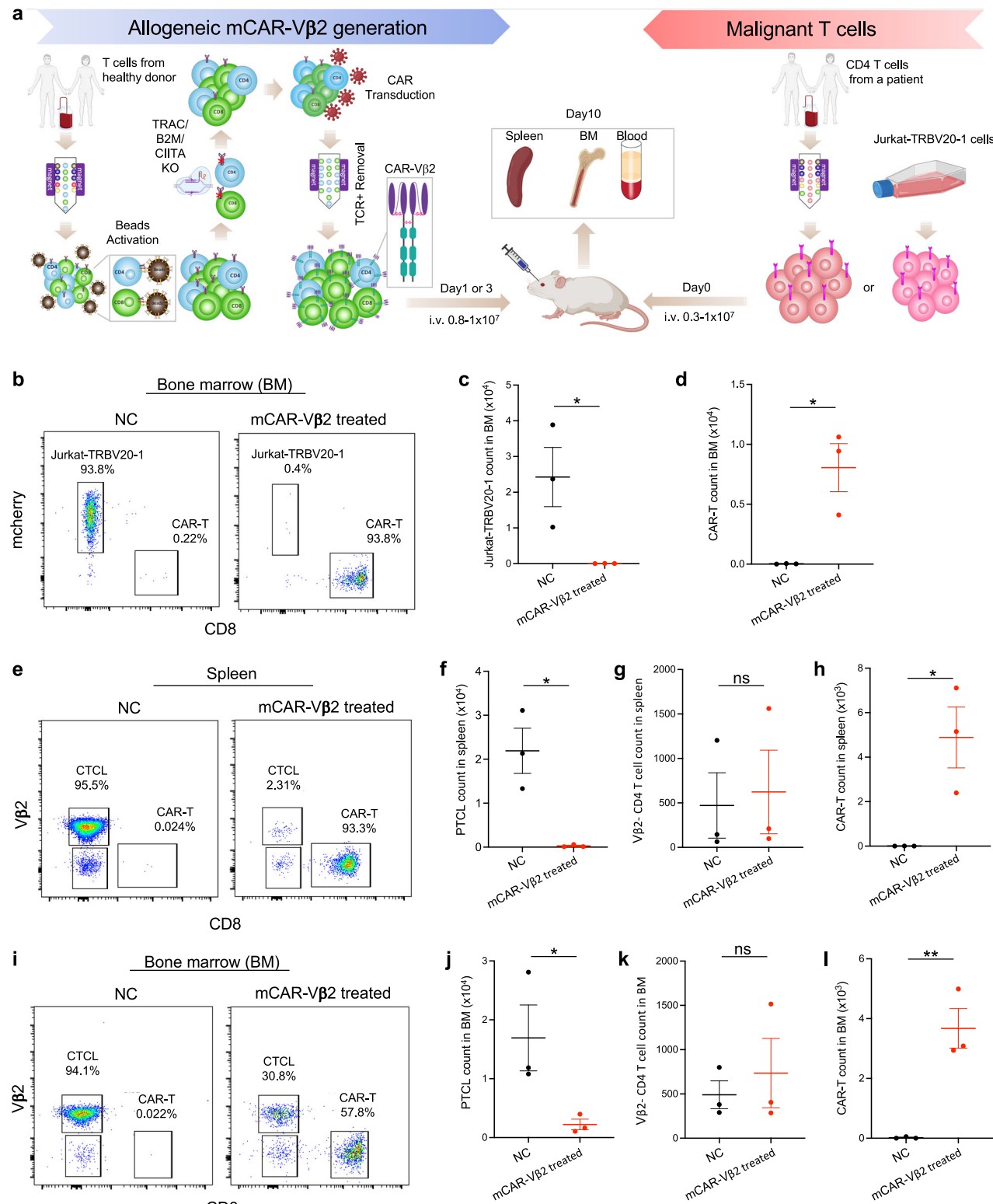

exogenous cytokine supplementation (Fig. 2b, d). Similarly, in our Vβ2+ PTCL patient PDX model, relative to control mice, mCAR-Vβ2 T cell treated mice consistently showed significantly reduced Vβ2+ PTCL cell counts in the spleen (Fig. 2e, f), bone marrow (Fig. 2i, j) and peripheral blood (Supplementary Fig. 2f), confirming killing efficacy against primary patient malignant T cells in vivo. Moreover, Vβ2-negative normal T cell counts showed no difference between control and mCAR-Vβ2 T cell treated groups across different sites including spleen (Fig. 2g), bone marrow (Fig. 2k) and peripheral blood

(Supplementary Fig. 2g), strongly suggesting that the killing was limited to Vβ2+ T cells and strictly CAR/ligand interaction dependent. Once again, the persistence of mCAR-Vβ2 T cells in vivo was also confirmed in spleen (Fig. 2e, h), bone marrow (Fig. 2i, l) and peripheral blood (Supplementary Fig. 2h) 1 week post intravenous transfusion without exogenous cytokine supplementation. These results reveal the efficient and specific capacity of allogeneic mCAR-Vβ2 T cells to kill in vivo not only an established Vβ2+ malignant T cell line but also primary Vβ2+ malignant T cells directly isolated from a patient.

**Fig. 2 | Efficient and specific elimination of Vβ2+ malignant T cells in vivo by allogeneic CAR-Vβ2 T cells. a** Schematic of the NSG mouse model used to assess in vivo Vβ2+ T lymphoma killing by mCAR-Vβ2 T cells. Illustrations were created with BioRender.com and Adobe Illustrator. **b–d** Analysis of NSG bone marrow (BM) 10 days post inoculation of Jurkat-TRBV20-1 cells and subsequent treatment with triple KO CD8 mCAR-Vβ2 T cells vs no-treatment control (NC). **b** Representative flow cytometry quantified in (**c**, **d**). **c** Vβ2+ Jurkat-TRBV20-1 cell count, and (**d**) mCAR-Vβ2 T cell count with (red) or without (black) mCAR-Vβ2 treatment. **e–l** Analysis of NSG mice carrying CD4+ T cells from a Vβ2+ PTCL patient 1 week post treatment with triple KO CD8 mCAR-Vβ2 T cells vs no-treatment control (NC).

**e** Representative NSG spleen cell flow cytometry quantified in (**f–h**) showing Vβ2+ PTCL cells, Vβ2-negative normal T cells and CD8 mCAR-Vβ2. **f** Vβ2+ PTCL cell count, (**g**) Vβ2-negative normal CD4+ T cell count, and (**h**) mCAR-Vβ2 T cell count in spleen with (red) or without (black) mCAR-Vβ2 treatment. **i** Representative NSG BM flow cytometry quantified in (**j–l**) showing Vβ2+ PTCL cells, Vβ2-negative normal T cells and CD8 mCAR-Vβ2. **j** Vβ2+ CTCL cell count (**k**) Vβ2- normal CD4+ T cell count, and (**l**) mCAR-Vβ2 T cell count in BM with (red) or without (black) mCAR-Vβ2 treatment. **c–l** N = 3 mice per group. Data are presented as mean values +/− SEM. *p < 0.05 and **p < 0.01 by two-sided t-test. Source data and exact p-values are provided as a Source Data file.

## Humanization of CAR-Vβ2 T cells with preserved efficacy and minimized immunoreactivity

CAR-T receptor humanization is a strategy shown to increase CAR-T survival after infusion[30,31]. Thus, to further modify mCAR-Vβ2 T cells for potential clinical use, we performed humanization of the mCAR-Vβ2 scFv domain by two different in silico strategies using purified CD8 mCAR-Vβ2 T cells. A total of 24 humanized CAR-Vβ2 (hCAR-Vβ2) candidates were generated: 12 using the BioPhi program designed to minimize T cell-dependent immunogenicity, and 12 by a third-party antibody humanization service (CDR grafting into selected human germline Vβ, performed by mAbvice). Eight of the hCAR-Vβ2 from BioPhi (Supplementary Fig. 3a) and all 12 (Supplementary Fig. 3b) from the third-party service showed preserved or improved in vitro killing of primary malignant T cells from a Vβ2+ CTCL patient when compared to mCAR-Vβ2 and (non-specific) CAR-CD19 controls. While lentiviral transduction efficiency was relatively even across different humanized versions as indicated by percent GFP+ (Supplementary Fig. 3c, e), CAR expression indicated as GFP mean fluorescent intensity (MFI) per cell varied substantially (Supplementary Fig. 3d, f), suggesting that the amino acid sequences of our humanized CARs may affect their protein stability and/ or expression efficiency. Moreover, all hCAR-Vβ2 T cell versions (Supplementary Fig. 4a) showed very limited pre-existing immunoreactivity to IgM in the sera of various healthy donors (Supplementary Fig. 4b) and CTCL patients (Supplementary Fig. 4c). To further assess the effectiveness of our triple-KO strategy for reducing immune reactivity, we utilized an in vitro mixed lymphocyte reaction (MLR) combining CD8+ T cells from multiple donors with allogeneic triple-KO hCAR-Vβ2 versus TRAC-KO hCAR-Vβ2 (Supplementary Fig. 4d). Together, these data suggest that hCAR-Vβ2 T cells would be tolerated after transfer to patients.

To study the in vivo killing by hCAR-Vβ2 T cells, we selected a top candidate, hCAR-Vβ2-(V7-4-1*02_1_V1-39*01_2; hereafter abbreviated simply as hCAR-Vβ2) based on the comprehensive criteria of higher percent GFP-positivity, GFP MFI, and in vitro killing efficiency; minimized pre-existing immunoreactivity to human serum; and higher protein L positivity that would be critical for hCAR-Vβ2 T cell pharmacological dynamic and kinetic studies in vivo. In a PDX NSG model with total CD4 T cells isolated from a Vβ2+ CTCL patient as the malignant T cell source and hCAR-Vβ2 T cells developed from pan T cells of a healthy donor, adoptively transferred human cells were quantified in bone marrow (Fig. 3a) and spleen (Fig. 3b) by flow cytometry. Relative to no-treatment (NC) and control CAR-CD19 T cell treated groups, the hCAR-Vβ2 treated group showed significantly reduced primary Vβ2+ CTCL cells (Fig. 3c, g) with an increased percentage of Vβ2-negative normal T cells (Fig. 3d, h) in both bone marrow (Fig. 3c, d) and spleen (Fig. 3g, h), demonstrating the high specificity and efficacy of hCAR-Vβ2 T cell killing of Vβ2+ malignant T cells in vivo. While the number of initially transferred CAR-T cells was the same between CAR-CD19 and hCAR-Vβ2 groups, endpoint CAR-T cell counts of both CD8 (Fig. 3e, i) and CD4 (Fig. 3f, j) subsets were significantly higher in the bone marrow (Fig. 3e, f) and spleen (Fig. 3i, j) of the hCAR-Vβ2 group, suggesting an activation-dependent improved CAR-T cell persistence in vivo. Overall, our CAR humanization strategies preserved the specific on-target recognition and killing efficacy both in vitro and in vivo while minimizing the potential risk of immunoreactivity in humans.

## CRISPR-AAV system as an alternative for CAR-Vβ2 engineering

Compared to lentiviral-dependent CAR expression that induces random genome integration, AAV-induced precise CAR genome integration combined with CRISPR KO may mitigate potential safety concerns of the lentiviral system for clinical use[32]. Thus, we designed an AAV construct containing a CAR-Vβ2 expressing cassette flanked by homology arms of the TRAC gene at the start region of exon 1 where a sgRNA of TRAC[28] was used to create a spCas9-induced double-stranded DNA break for AAV template DNA integration thereafter (Fig. 4a). The beginning of the CAR-expressing cassette was designed to be in frame with original TCR alpha chain translation and then separated by a T2A self-cut peptide for simultaneous endogenous TCR KO and CAR-Vβ2 expression under endogenous TCR alpha promoter control[28]. To test our CRISPR-AAV system, we used mCAR-Vβ2 and detected its expression on pan T cells from a healthy donor, showing a substantial and more uniform CAR-Vβ2 expression level (reported as GFP) on both CD4 and CD8 T cells (Supplementary Fig. 5a). In addition, percent CAR-Vβ2 positivity was modestly increased during in vitro cytokine expansion culture from about 60% at day 2 to about 70% at day 10 post AAV transduction (Supplementary Fig. 5b). In vitro killing efficacy of AAV transduced mCAR-Vβ2 T cells was confirmed in co-culture with Vβ2+ CTCL cells from a patient, and compared to CAR-CD19 T cells as a negative control (Supplementary Fig. 5c) demonstrating the feasibility of the CRISPR-AAV system for CAR-Vβ2 T cell generation. We further produced hCAR-Vβ2 T cells by the same method, obtaining consistently high and uniform CAR expression level, although the percent CAR positivity of AAV-transduced cells was slightly lower than that of lentivirus-transduced ones (Fig. 4b). Compared to CAR-CD19 T cells, AAV-transduced hCAR-Vβ2 T cells showed effective in vitro killing of primary Vβ2+ malignant T cells from two different patients (Fig. 4c). Furthermore, Jurkat-TRBV20-1 (Vβ2+) cells but not Jurkat-TRBV6-2 (Vβ13.2+) cells were significantly killed by AAV-transduced hCAR-Vβ2 T cells (Fig. 4d), indicating the specific recognition of hCAR-Vβ2. We next assessed the in vivo efficacy of AAV-transduced hCAR-Vβ2 T cells using PDX NSG mice prepared by transferring primary total CD4 T cells isolated from a Vβ2+ PTCL patient. Compared to no-treatment (NC) and CAR-CD19 treated groups, the AAV-transduced hCAR-Vβ2 group showed significantly reduced Vβ2+ malignant T cell counts in spleen (Fig. 4e) and bone marrow (Fig. 4i), while Vβ2-negative normal T cell counts were not different (Fig. 4f), confirming the specific killing by AAV-transduced hCAR-Vβ2 T cells of Vβ2+ malignant T cells without affecting normal T cell viability. CD69 was significantly increased in both CD8 (Fig. 4g, j) and CD4 (Fig. 4h, k) subsets of AAV-transduced hCAR-Vβ2 T cells in spleen (Fig. 4g, h) and bone marrow (Fig. 4j, k) when compared to those of CAR-CD19 T cells, also supporting the activation of AAV-transduced hCAR-Vβ2 T cells in vivo and correlated to the observed increased Vβ2+ malignant T cell killing. While the PDX model established with primary malignant T cells from patients may better reflect the potential immediate hCAR-Vβ2 T cell killing in the clinic, we found that it may not be optimal for the study of long-term and persistent CAR-T responses as transferred patient primary malignant T cells do not persist in NSG mice for a long time. Therefore, we generated a stable luciferase-expressing Jurkat-TRBV20-1 (Jurkat-TRBV20-1-lucifer) clone to establish a long-term PDX model.

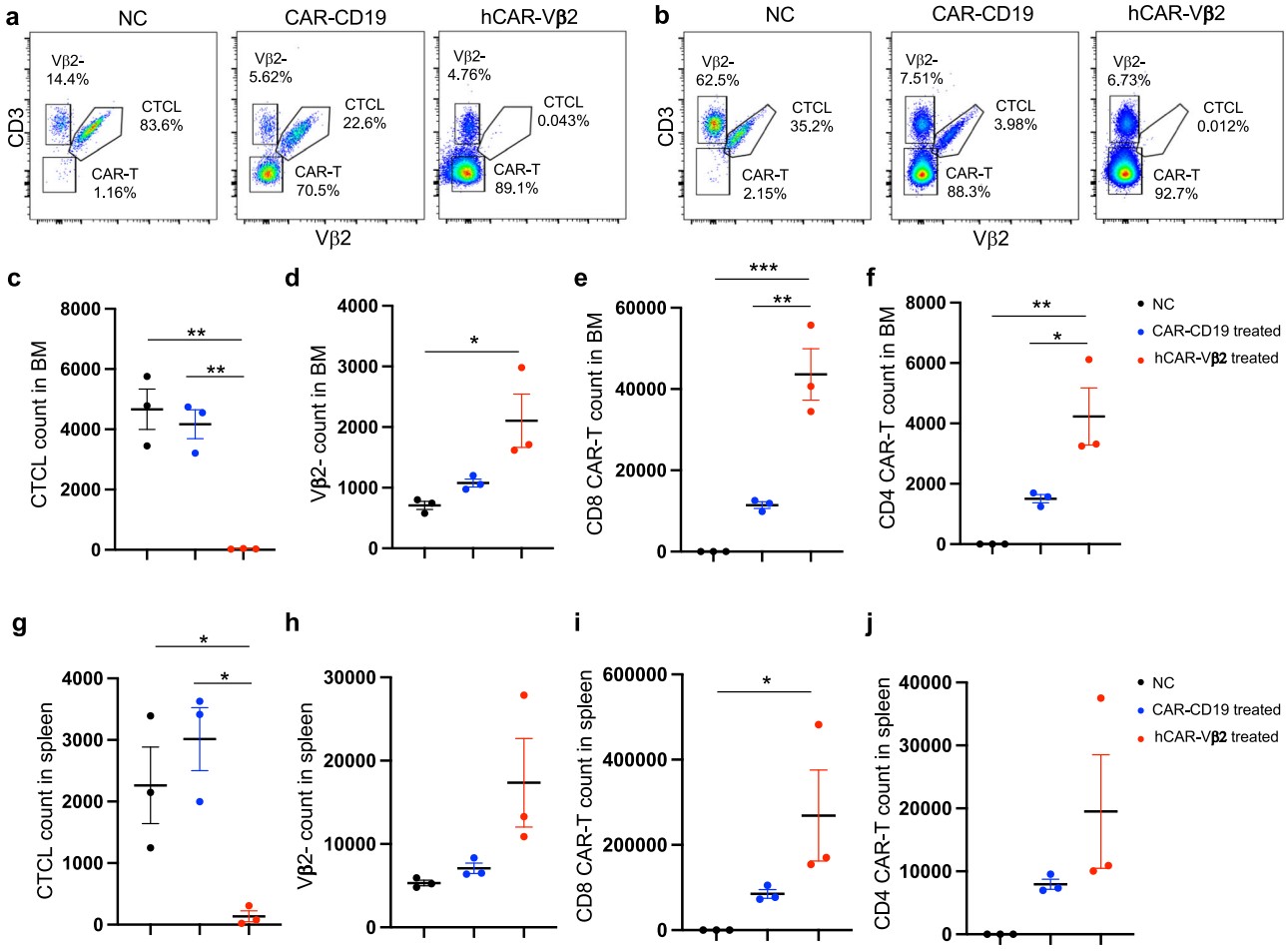

**Fig. 3 | Humanization of CAR-Vβ2 T cells with preserved efficacy.**
**a, b** Representative flow cytometry of (**a**) BM and (**b**) spleen, showing CD3+ Vβ2+ CTCL cells, CD3+ Vβ2-negative normal T cells and CD3-negative hCAR-Vβ2 T cells 3 days after in vivo hCAR-Vβ2 treatment, nonspecific CAR-CD19 treatment, or no-treatment control (NC). **c** Vβ2+ CTCL cells (**d**) Vβ2-negative normal T cells (**e**) CD8 hCAR-T and (**f**) CD4 hCAR-T cells in NSG BM 3 days after in vivo allogeneic triple-KO

hCAR-Vβ2 (red) or CAR-CD19 (blue) pan-T cell treatment generated from a healthy donor compared to no-treatment control (NC, black). **g** Vβ2+ CTCL cell counts (**h**) Vβ2-negative normal T cell counts (**i**) CD8 CAR-T cell counts and (**j**) CD4 CAR-T cell count in spleen of the same mice. **c–j** $N = 3$ mice per group. Data are presented as mean values +/− SEM. $*p < 0.05$, $**p < 0.01$ and $***p < 0.001$ by one-way ANOVA. Source data and exact $p$-values are provided as a Source Data file.

Lentiviral-transduced CAR-CD19 (lenti CAR-CD19) T cells, lentiviral-transduced hCAR-Vβ2 (lenti hCAR-Vβ2) T cells and AAV-transduced hCAR-Vβ2 (AAV hCAR-Vβ2) T cells were generated from triple-KO pan T cells of a healthy donor, with similar CAR expression level confirmed by flow cytometry detection of GFP (Supplementary Fig. 5d). Weekly live bioluminescent imaging of both ventral (Fig. 4l) and dorsal positions (Fig. 4m) of NSG mice populated with Jurkat-TRBV20-1-lucifer cells, revealed that both lenti hCAR-Vβ2 and AAV hCAR-Vβ2 groups showed markedly lower signals over 30 days compared to no-treatment and lenti CAR-CD19 groups (Fig. 4n, o). Consistently, there was no significant body weight loss in lenti CAR-Vβ2 and AAV CAR-Vβ2 groups over time (Fig. 4p) and a significant survival advantage of these two treatment groups (Fig. 4q). Thus, allogeneic hCAR-Vβ2 T cells generated by either the lentivirus or AAV system showed long-term efficacy against Vβ2+ malignant T cell disease in PDX mice without detectable safety issues in vivo.

## Cytokine combinations for allogeneic hCAR-Vβ2 T cell expansion and differentiation
To consider potential culture conditions for ex vivo allogeneic hCAR-Vβ2 T cell production, we explored several cytokine combinations and their effects on CAR-T expansion, differentiation, and killing. Several cytokines, such as IL2, IL7, IL12, IL15, IL18 and IL21, associated

with effector and/or memory T cell differentiation, survival, and proliferation have been investigated in CAR-T cell generation[33–41]. To systematically examine the culture conditions for hCAR-Vβ2 T cell production, we screened 20 different cytokine combinations (Supplementary Table 1) to evaluate hCAR-Vβ2 T cell survival, expansion, acute and repeated killing ability, relevant cytokine production and effector/memory surface markers over 14 days in vitro (Fig. 5a). A subset of 7 of these conditions were used to confirm CAR-T cell expansion and chronic/repeated killing capacity using CAR-T prepared from two additional healthy donors. Cells in the no cytokine control did not expand (Supplementary Fig. 6c), consistent with a cytokine-dependent expansion of our hCAR-Vβ2 T cells. In all other groups, the CD8+ subpopulation gradually expanded and became dominant in hCAR-Vβ2 T cells at day 14 (Supplementary Fig. 6a), with an increased percentage of CD45RA + CD45RO+ (Supplementary Fig. 6b). Combinations IL7 + IL15 + IL12, IL7 + IL15 + IL21 and IL7 + IL15 + IL12 + IL21 promoted T cell expansion across three donors. In contrast, combinations containing IL12 + IL18 were observed to reduce T cell expansion (Fig. 5b and Supplementary Fig. 6c) and viability (Supplementary Fig. 6d) in correlation with higher levels of IFNγ expression in culture without CAR engagement (Supplementary Fig. 6e), suggesting an over-activation of T cells under IL12 + IL18 conditions.

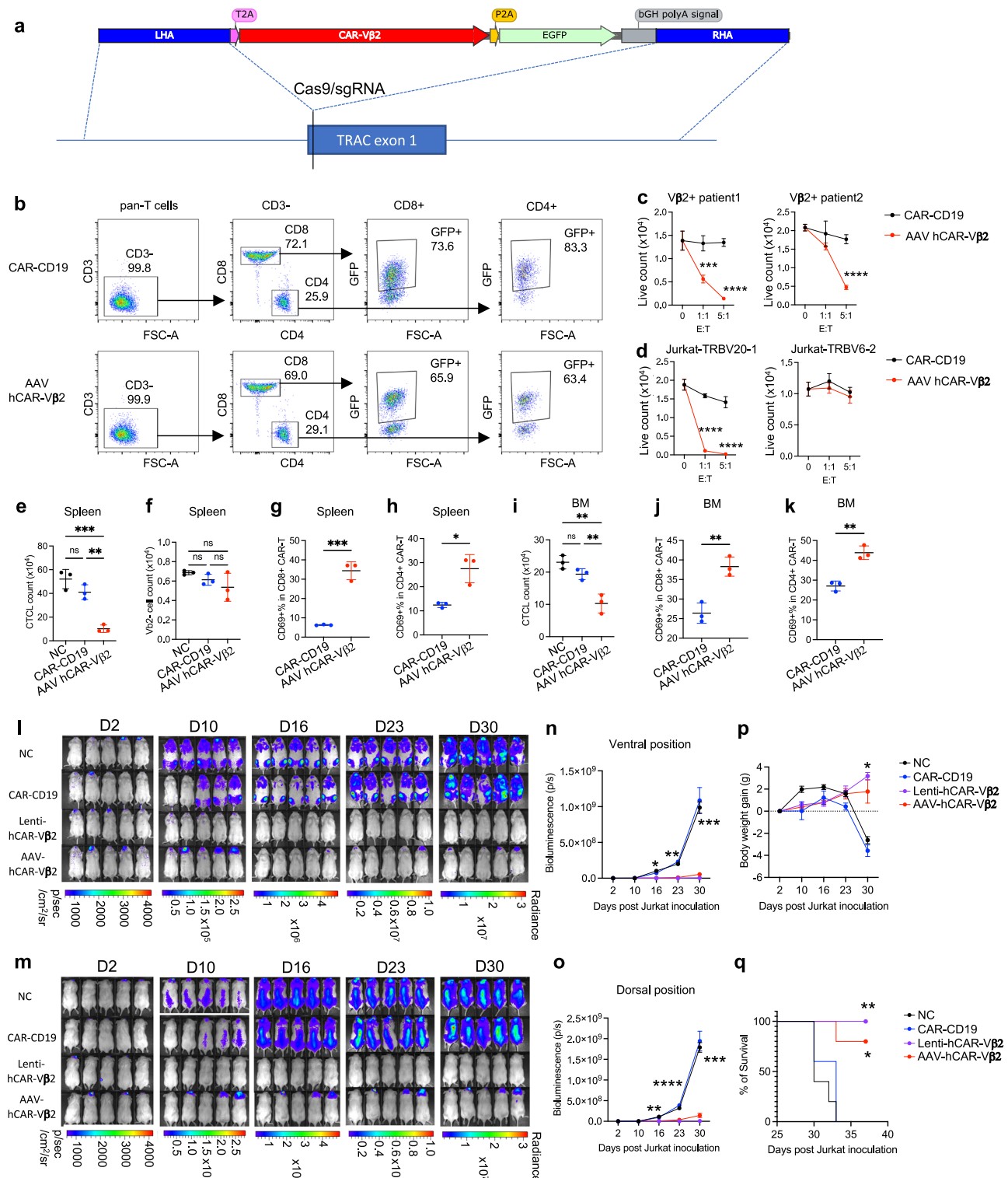

Acute repeated Jurkat-TRBV20-1 cell killing by day 9 expanded hCAR-Vβ2 T cells was next evaluated for 4 consecutive days, with gradually increasing numbers of live Jurkat-TRBV20-1 target cells added at day 0, 1, and 2, without exogenous cytokine supplementation. IL12-containing conditions maintained strong killing and CAR-T cell re-expansion, again in association with an increased CD45RA + CD45RO+ percentage within the CD8+ subpopulation, while other conditions resulted in gradually reduced killing and re-expansion capacity (Fig. 5c–e). Notably, IL21 failed to maintain killing ability even

though it promoted TNFα (Supplementary Fig. 6f) and GZMB (Supplementary Fig. 6g) expression by hCAR-Vβ2 T cells in culture without CAR engagement, the effect of which can nonetheless be suppressed by IL12. To also assess the effects of chronic stimulation, hCAR-Vβ2 T cells generated from two additional donors were stimulated with Jurkat-TRBV20-1 cells every 3 days, with killing assessed after each cycle. hCAR-Vβ2 T cells generated in IL7 + IL15 + IL12 consistently sustained Jurkat-TRBV20-1 cell killing ability (Fig. 5f), although other cytokine conditions were also supportive (summarized in Fig. 5g).

**Fig. 4 | CRISPR-AAV system for CAR-Vβ2 T cell generation and treatment of Vβ2+ PTCL PDX. a** Adeno-associated virus (AAV) chimeric antigen-receptor (CAR) template structure and the strategy to integrate into the *TRAC* region using Cas9-*TRAC*-sgRNA. **b** Representative flow cytometry showing CD3- purity, CD4 and CD8 population percentages and CAR expression of AAV-dependent allogeneic hCAR-Vβ2 T cells compared to lentiviral-dependent allogeneic CAR-CD19 T cells. **c** Live PTCL/CTCL cell counts from two Vβ2+ patients and (**d**) live Jurkat-TRBV20-1 (Vβ2+, left) or Jurkat-TRBV6-2 (Vβ13.2+, right) cell counts after overnight in vitro culture with allogeneic lenti-CAR-CD19 T cells (black) or AAV-hCAR-Vβ2 T cells (red) at different E:T ratios, determined by flow cytometry. **e–k** Total CD4 T cells isolated from a Vβ2+ PTCL patient were adoptively transferred into groups of NSG mice that were then treated with allogeneic triple-KO AAV-hCAR-Vβ2 (red) or lenti-CAR-CD19 (blue) generated from healthy donor pan T cells, compared to no-treatment control (NC, black). Three days post-treatment (**e**) Vβ2+ CTCL cells (**f**) Vβ2- normal T cells

(**g**) CD69+ % in CD8 CAR-T cells, and (**h**) CD69+ % in CD4 CAR-T cells in spleen were quantified by flow cytometry, as were (**i**) Vβ2+ CTCL cells (**j**) CD69+ % in CD8 CAR-T cells, and (**k**) CD69+ % in CD4 CAR-T cells in BM. **l–o** Long-term bioluminescence monitoring (IVIS) of Jurkat-TRBV20-1-lucifer cell-bearing NSG mice, at (**l–n**) ventral or (**m–o**) dorsal position, without treatment (NC, black) or following treatment with CAR-CD19 T cells (blue), lenti-hCAR-Vβ2 T cells (purple) or AAV-hCAR-Vβ2 T cells (red). **p**, Body weight and (**q**) survival of the same mice. **c**, **d** *n* = 3 replicates in each group. ***$p < 0.001$ and ****$p < 0.0001$ by two-way ANOVA. **e-k**, *n* = 3 mice in each group. *$p < 0.05$, **$p < 0.01$ and ***$p < 0.001$ by one-way ANOVA. **n–q** *n* = 5 mice in each group. **n**, **o** *$p < 0.05$, **$p < 0.01$, ***$p < 0.001$ and ****$p < 0.0001$ by two-way ANOVA. **q** *$p < 0.05$ and **$p < 0.01$ by survival analysis (Kaplan-Meier). **c–k–n–q** Data are presented as mean values +/− SEM. All replicates are independent samples. Source data and exact *p*-values are provided as a Source Data file.

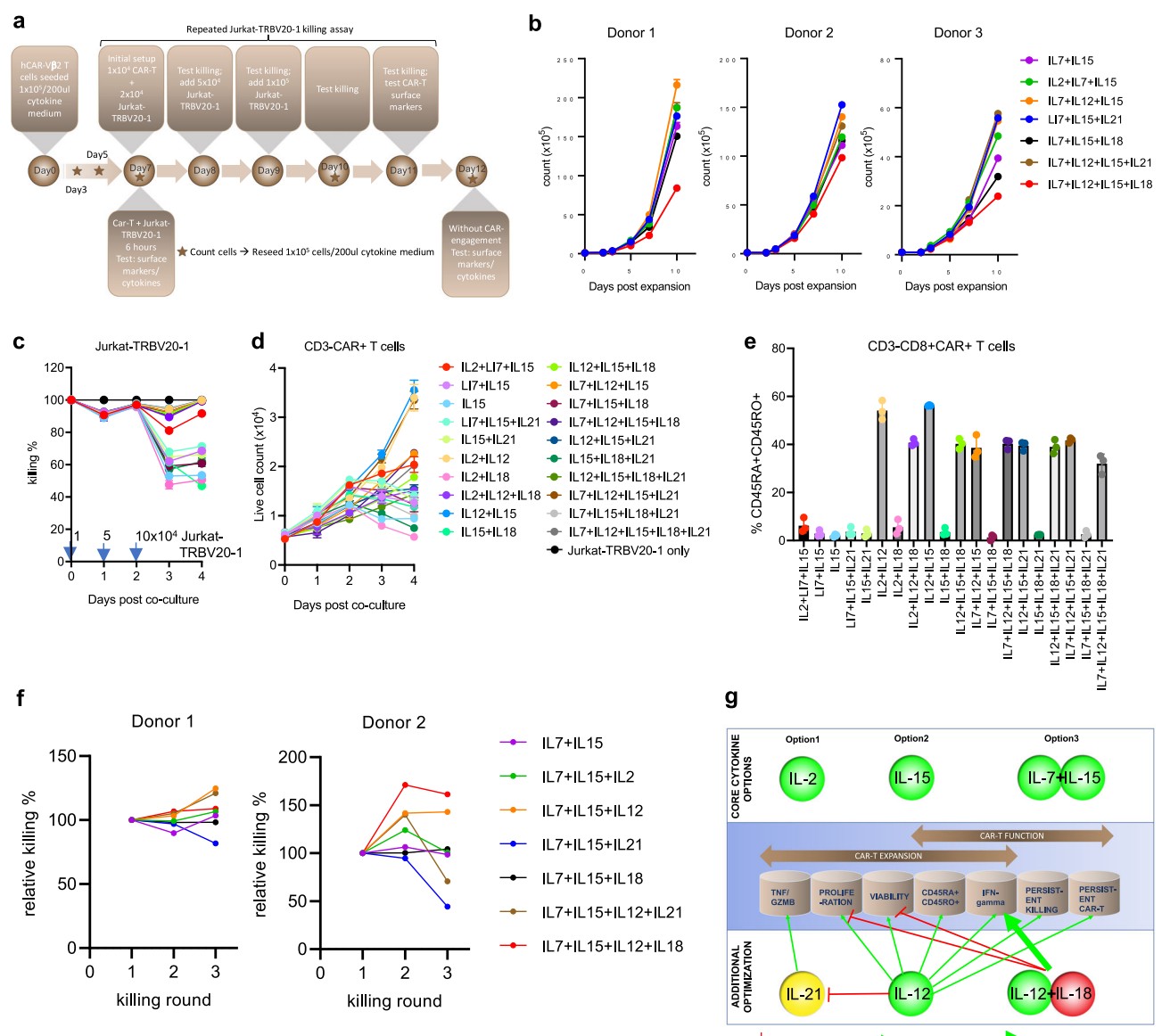

**Fig. 5 | Cytokine optimization for hCAR-Vβ2 T cell expansion in vitro.**
**a** Schematic of the timeline of cytokine optimization and functional analysis. **b** Live total T cell counts during in vitro expansion of hCAR-Vβ2 T cells from three different donors, determined by trypan blue cell counting. **c** Jurkat-TRBV20-1 killing % and (**d**) live CD3- hCAR-Vβ2 T cell counts in acute repeated Jurkat-TRBV20-1 killing assay, determined by trypan blue cell counting and flow cytometry. **e** CD45RA+ CD45RO+ % of CD8 + CD3- hCAR-Vβ2 T cells 4 days post repeated Jurkat-TRBV20-1 cell killing assay, determined by flow cytometry. **b–e** *N* = 3

replicates in each group. Data are presented as mean values +/− SEM. **f** relative Jurkat-TRBV20-1 killing by chronically stimulated hCAR-Vβ2 T cells from two healthy donors generated in seven cytokine conditions, with each killing round normalized to the first round. **g** summary of optimized cytokine combinations for allogeneic hCAR-Vβ2 T cell expansion in vitro. Illustrations in Fig. 5a, g were created using Adobe Illustrator. All replicates are independent samples. Source data are provided as a Source Data file.

### Humanized anti-Vβ2 IgG1 antibody with enhanced ADCC as another therapeutic option for Vβ2+ T cell malignancy

To expand the potential for anti-Vβ therapeutics to treat patients with personalized agents matched not only to their Vβ expression, but as well to specific clinical settings (i.e. different TCL types and stages may be better suited for treatment with a therapeutic antibody versus a CAR-T), we also developed an allogeneic NK-cell based ADCC platform. As is generally known, patients with T cell malignancies often show increased opportunistic infections due to compromises in normal T cell and NK numbers and/or function[42–55]. We reasoned that allogeneic T cells with their more efficient transduction efficiency (relative to NK cell transduction) are suitable for CAR engineering directly, while donor NK cells could utilize infused therapeutic antibody via their Fc receptors as an alternative and complimentary therapeutic strategy. Thus, we generated humanized anti-Vβ2 (h-anti-Vβ2) IgG1 antibodies with enhancing ADCC mutations at the Fc region[56]. In addition, the focosyl transferase 8 (Fut8) gene was knocked out of the antibody producing cell line, expiCHO, to further enhance ADCC functionality by eliminating post-translational fucosylation of produced antibodies[57] (Supplementary Fig. 7a). The purity of generated h-anti-Vβ2 post protein G purification and PBS dialysis was confirmed by SDS-PAGE (Supplementary Fig. 7b). Competitive binding of h-anti-Vβ2 and original mouse anti-Vβ2 (m-anti-Vβ2) during co-staining of Jurkat-TRBV20-1 cells showed that the affinity of h-anti-Vβ2 to TRBV20-1 antigen was similar to that of m-anti-Vβ2 (Fig. 6a). Because ADCC is one major effect of therapeutic antibodies, we firstly assessed h-anti-Vβ2 ADCC activity in vitro by using Jurkat-TRBV20-1 cells as target cells. Compared to m-anti-Vβ2, Fc optimized h-anti-Vb2 IgG1 showed a significantly higher ADCC effector signal, with a saturated responding concentration of 100 ng/ml (Fig. 6b). An ADCC response of h-anti-Vβ2 IgG1 mediated by NK cells isolated from a healthy donor was also demonstrated against Jurkat-TRBV20-1 cells (Supplementary Fig. 7c). To confirm the ADCC effect against primary malignant T cells from patients, PBMCs from two different Vβ2+ CTCL patients were co-cultured with NK cells from a healthy donor. Compared to no antibody and m-anti-Vβ2 antibody control groups, h-anti-Vβ2 group showed more efficient Vβ2+ CTCL killing (Fig. 6c, f), without influencing the viability of Vβ2-negative normal T cells (Fig. 6d, g) and non-T cells (Fig. 6e, h). Although further preclinical development is necessary to fully optimize allogeneic hCAR-Vβ2 T cells and ADCC-enhanced and humanized anti-Vβ2 IgG1 antibodies before assessment in clinical trials, the strategies and protocols utilized herein may serve as an efficient platform for the generation of such promising Vβ-targeting immunotherapeutic agents.

## Discussion

Currently investigated immunotherapies for T cell malignancy include therapeutic antibodies, bispecific antibody-T cell engagers, and CAR-T cells targeting CD5, CD7, CD30, CD37, CCR4, TRBC1 (refs. 1,7; and clinicaltrials.gov). Many challenges exist for these strategies[58–60] including relapse after antibody treatment, malignant T cell masking of bispecific antibody, shared expression of targeted surface markers between malignant T cells and normal T cells leading to CAR-T cell fratricide and T cell aplasia induced immune suppression, and resistance to CAR-T therapy by contamination of CAR-T cell products with malignant T cells, similar to that described in a leukemic B-ALL case[61]. In these patients, malignant CD19-CAR+ cells bound to CD19, thus preventing recognition by functional CAR-T cells. A promising CAR-T approach to T cell malignancy is the targeting of one of the two potential T cell receptor beta constant regions, TRBC1 or TRBC2. Maciocia et al. have shown that the proportion of TRBC1 + T cells varies between 25 and 47% in healthy donors, regardless of the T cell subset[62]. T cell leukemias and lymphomas, instead, are either clonally TRBC1 positive or negative. Therefore, TRBC1 CAR-T cells may specifically eliminate TRBC1+ malignancies and normal T cells while sparing

TRBC2+ normal T-cells. A clinical trial testing TRBC1 CAR-T cells in T cell lymphomas is ongoing (AUTO4). Even so, this approach will result in substantial TRBC1+ normal T cell depletion and it is as yet unclear whether the residual T cell repertoire will be sufficient to maintain defense against pathogens and/or cancer cells.

To overcome these potential drawbacks and circumvent potential risks, we sought to target the single specific TCR-Vβ expressed on each T malignancy. A similar strategy was recently reported[63] showing that TCR-Vβ can be a target for CAR-T development, but the alloreactivity of CAR-T from healthy donors to patients, autologous CAR-T efficiency and TRBV-directed CAR-T efficacy against primary malignant T cells directly isolated from patients were not fully investigated, making the translational potential of this strategy uncertain. In our study, we demonstrated that the application potential of autologous CD8 CAR-Vβ2 T cells was low and inefficient. In contrast, allogeneic CD8 or pan CAR-Vβ2 T cells generated from healthy donors were highly efficient in on-target killing of Vβ2+ malignant T cells directly isolated from CTCL/PTCL patients, with demonstrated specificity and safety when endogenous TCRs are knocked out to minimize potential GVH activity.

We further performed MHC-I/II KO and CAR humanization to minimize the potential HVG effects to allogeneic CAR-Vβ2 T cells and confirmed reduced immune reactivity (Supplementary Fig. 4d), suggesting that our allogeneic triple-KO CAR-T cells may show persistence in treated patients. This would be especially likely given that T cell malignancy patients show T cell functional deficiencies and CAR-T protocols often require induction chemotherapy that would further compromise T cell-mediated immunity against the CAR-T. As the functions of normal T cells are often compromised in T malignancies, it is highly possible that development of high affinity, effective IgG against allogeneic humanized CAR-T cells, which typically requires T cell helper function, may be compromised as well, although CAR-T treatment may prove to normalize T cell responses after significantly lowering the malignant T cell burden. Such may also be the case in recipient NK cell-dependent allogeneic MHC-I KO CAR-T cell rejection. Several prior studies have shown that further manipulation of certain genes (e.g. CD47 or HLA-E) in allogeneic CAR-T cells was able to significantly inhibit NK-dependent rejection[64–66], which can be incorporated into future iterations of allogeneic CAR-Vβ T cells for enhanced persistence in patients. Whether potential rejection of our engineered allogeneic CAR-Vβ T cells is critical or not for the treatment efficacy and safety is unknown. Since the sgRNAs targeting *TRAC*, *B2M* and *CIITA* in these studies are only used for proof-of-principle, the potential off-target effects would need to be carefully screened prior to clinical product development. Optimization of such might include using high-fidelity Cas9 protein, screening multiple *B2M* and *CIITA* sgRNAs, and adjusting electroporation parameters.

Herein, we also developed both lentiviral- and AAV-dependent stable CAR-expressing systems, showing similar transduction efficiency and malignant T cell control efficacy. While the majority of current CAR-T products on the market and in clinical trials are based on lentiviral transduction, concerns remain that the random genome integration properties of the lentiviral system may induce genetic mutations that might increase the chance of malignant transformation of CAR-T cells[67]. However, whether (and to what degree) lentiviral mediated integration increases the risk of leukemic transformation in human CAR-T or other T cells will require long-term follow-up clinical studies. The sgRNA-guided site-specific integration of CAR constructs in the AAV system may prove to further minimize this potential safety concern while maintaining or even enhancing CAR-T efficacy.

Cytokines as the third signal for optimal T cell priming and effector function are critical for CAR-T cell product expansion in vitro. Although initially IL2 was routinely used alone in CAR-T cell expansion procedures, it was later demonstrated to promote CAR-T cell terminal effector and exhausted phenotypes without persistent in vivo tumor killing effects[36]. Thereafter, IL7 and IL15 were shown to improve

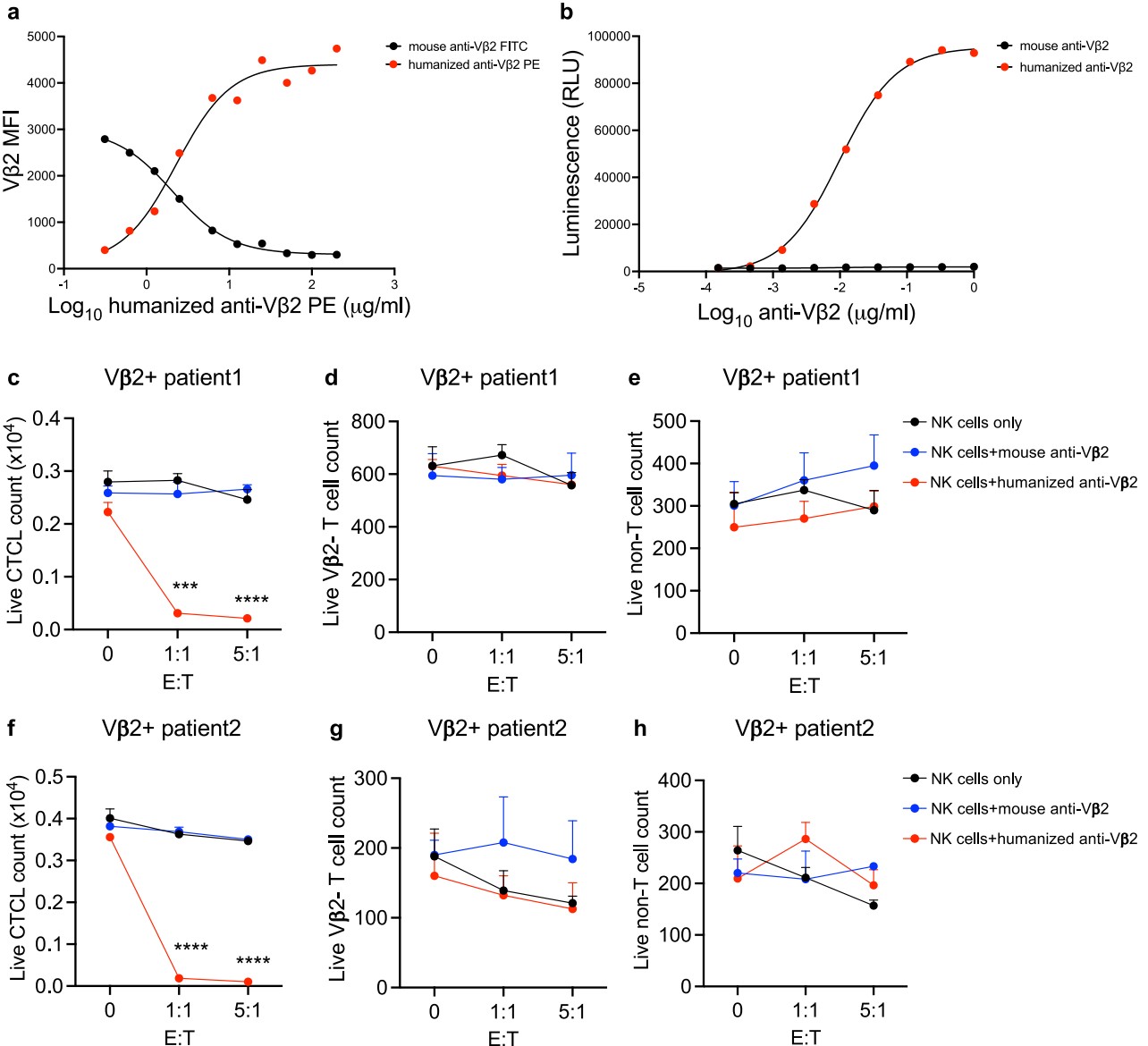

**Fig. 6 | Humanized anti-Vβ2 therapeutic antibody with enhanced ADCC. a** An anti-Vβ2 antibody competition assay, in which 1.25μg/ml mouse anti-Vβ2-FITC antibody was mixed with varying concentrations of humanized anti-Vβ2-PE antibody and used to stain CTCL cells from a Vβ2+ patient, with Vβ2 MFI determined by flow cytometry. **b** ADCC assay luminescence signal after a 6 h co-culture of patient derived Vβ2+ target CTCL cells and Jurkat-NFAT-luciferase effector cells mixed with either 100 ng/ml mouse anti-Vβ2 antibody (black) or humanized anti-Vβ2 antibody (red). PBMC from Vβ2+ CTCL patient 1 (**c**–**e**) or from Vβ2+ CTCL patient 2 (**f**–**h**) were cultured overnight with NK effector cells from a healthy donor at different E:T ratios without antibody addition (black) or with addition of mouse anti-Vβ2 antibody (blue) or humanized anti-Vβ2 antibody (red), then live CTCL cells (**c**–**f**) Vβ2-normal T cells (**d**–**g**) and non-T cell PBMC (**e**–**h**) counts were determined by flow cytometry. **c**–**h** $N = 3$ replicates in each group. Data are presented as mean values +/− SEM. ***$p < 0.001$ and ****$p < 0.0001$ by two-way ANOVA. All replicates are independent samples. Source data and exact $p$-values are provided as a Source Data file.

preservation of desirable stemness and memory phenotypes[36]. Furthermore, IL12[33,39,41], IL18[34,38] and IL21[37] are shown to have distinct influences in shaping T cell priming and later effective responses. Some studies have compared the effects of several different limited cytokine combinations on CAR-T development[37,40], but these have not been comprehensive. Herein, we conducted a systematic comparison with a more comprehensive range of relevant cytokine-cocktail conditions based on functional group combinations. We included IL2, IL7, IL12, IL15, IL18 and IL21 together and subdivided them into three groups: an effector phenotype promoting group (containing IL2, IL12 and IL18), a memory phenotype promoting group (containing IL7 and IL15), and a hybrid effect group containing IL21. Individual groups or combinations of these different groups formed the cytokine-cocktail conditions to screen for the optimal CAR-T conditions regarding the

level of product expansion, acute and persistent target cell killing efficacies in vitro, and the expansion/persistence of CAR-T cells during CAR engagement. We found that IL12 + IL2, alone or combined with memory-promoting IL7/IL15, showed superior effects regarding all criteria. Without IL12, IL7 and IL15 can maintain a memory/naïve phenotype of CAR-T cells but failed to induce persistent effective killing responses and expansion upon CAR engagement. IL18 has been shown to be a T cell response booster[34,38]; however, without IL12, IL18 alone did not show significant beneficial effects in combination with other cytokines. On the contrary, when IL18 was used in combination with IL12, CAR-T cells showed strong activation even at resting status without CAR engagement indicated by production of high amounts of IFNγ. Although IL12 + IL18 resulted in superior persistent killing by CAR-T cells, this condition also induced significantly higher CAR-T cell

death and a lower expansion rate, probably due to the over-activation induced terminal effector differentiation and apoptosis. IL21 has also been shown to enhance T cell anti-tumor responses[37]. Indeed, we noticed that IL21 can induce higher GZMB and TNFα but not IFNγ production in CAR-T cells, and IL12 seems to counteract this effect. Nonetheless, we did not observe significant improvement of our CAR-T product regarding the criteria. This may be due to which cytotoxic mechanism the targeted malignant cells are more sensitive to, so we cannot exclude the possible beneficial effects of IL21 to CAR-T cell development in other contexts.

In addition to a CAR-T strategy, antibody therapies have been developed to target T malignancies, including mogamulizumab for CCR4, brentuximab vedotin for CD30, alemtuzumab for CD52, daratumumab and isatuximab for CD38, and camidanlumab tesirine for CD25[8]. Although antibody alone can direct the killing of malignant cells by endogenous NK cells, myeloid cells, and the complement system, such may not reach optimal levels due to the immunocompromised and immunosuppressive environment established chronically in patients, either directly by the malignant T cell burden and/or by prior lymphodepletion or immune inhibiting treatments. Therefore, in this study we also developed a healthy donor-derived allogeneic NK cell-dependent anti-Vβ ADCC therapy strategy, aiming to consistently and specifically kill malignant T cells by optimizing the potential of anti-Vβ therapeutic antibodies. With the progress of iPSC-derived NK cell production[68] and significantly improved feeder-free mature NK cell expansion techniques[69], the shortage source issue of functional allogeneic NK cells may be gradually overcome, making allogeneic NK cell-dependent ADCC therapy practical and affordable. We believe that for T cell malignancies that often show increased opportunistic infections with normal T cell and NK functional compromise, both strategies are viable and appropriate. Donor (allogeneic) T cells with their more efficient transduction efficiency (relative to NK cell transduction) are suitable for CAR engineering directly, while donor NK cells can utilize infused therapeutic antibody via their Fc receptors as an alternative strategy.

Relative to other investigated antigen targets on malignant T cells, a unique TCR-Vβ is stringently expressed by each patient's clonal malignant T cells. The potential drawback of applying this therapeutic approach more broadly, however, is the wide diversity of anti-TCR-Vβ CAR and/or antibody agents that would be necessary to potentially cover all possible Vβ isotypes. Towards that ultimate goal, we are developing a full panel of humanized anti-Vβ therapeutic antibodies and CAR agents. Nonetheless, the diversity of TCR-Vβ usage by malignant T cells is somewhat skewed towards a more limited potential number that can make anti-Vβ-based therapies feasible and applicable to the majority of T malignancy patients under a uniform production pipeline as demonstrated herein. Lastly, it has not escaped our notice that the specific grouping of anti-Vβ therapeutic agents may be individually matched to the most highly pathogenic and oligoclonally expanded T cells mediating autoimmune diseases[18,19]. In this way, high-throughput single-cell sequencing and/or multispectral imaging screening strategies may 1 day define the optimal combination of anti-Vβ therapeutic agents to most precisely treat each individual's T-cell mediated autoimmune disease, while minimizing effects on normal T cells. In summary, we comprehensively investigated and demonstrated the therapeutic potential of TCR-Vβ chain targeting immunotherapies via off-the-shelf allogeneic humanized CAR-Vβ T cells or allogeneic NK cell-dependent humanized anti-Vβ ADCC agents, providing a promising treatment strategy for T malignancies, and potentially as well for T cell-mediated autoimmune disorders.

## Methods

This research complies with all relevant ethical regulations and has been approved by the Yale Institutional Review Board, Human Investigation Committee and Institutional Animal Care and Use Committee.

### Human samples and cell lines

Peripheral blood from CTCL patients and healthy donors was collected in lithium heparin tubes at the Yale Cancer Center after obtaining written informed consent and following all ethical regulations in accordance with the Yale Human Investigational Review Board. The sex of donors was not considered in the study design. Individuals of both sexes (assigned at birth from their medical record), regardless of gender, were invited to participate in the study. Low sample size precludes sex- and gender-based analyses. Jurkat (ATCC, TIB-152, Clone E6-1) was purchased from ATCC and sub-cultured according to ATCC instructions. ExpiCHO cell line was from the ExpiCHO expression system kit (Thermofisher, A29129) and Jurkat-NFAT-luciferase effector cells was from ADCC Reporter Bioassay, Core Kit (Promega, G7010). They are sub-cultured based on manufacture's instruction.

### Animals

Eight to twelve weeks NOD.Cg-$Prkdc^{scid}$ $Il2rg^{tm1Wjl}$/SzJ (NSG) mice were purchased from The Jackson Laboratory. All of the in vivo studies were approved by the Yale Institutional Animal Care and Use Committee (IACUC). Mice were bred and maintained under specific pathogen-free conditions with a 12 hr light/dark cycle at 70–72 °F, 45–50% humidity with food and water provided ad libitum. The Yale animal facility is accredited by the Association for Assessment of Laboratory Animal Care. Experiments utilized both male and female mice that were age and sex-matched for each experiment. In accordance with Yale IACUC tumor burden is assessed and limited to that resulting in a body condition score < 2 and/or weight loss > 15%.

### Plasmids

pSLCAR-CD19-BBz was a gift from Scott McComb (Addgene plasmid # 135992; http://n2t.net/addgene:135992; RRID:Addgene_135992)[20], pAW13.lentiguide.mCherry was a gift from Richard Young (Addgene plasmid # 104375; http://n2t.net/addgene:104375; RRID:Addgene_104375)[70], pAAV.CMV.PI.EGFP.WPRE.bGH was a gift from James M. Wilson (Addgene plasmid # 105530; http://n2t.net/addgene:105530; RRID:Addgene_105530), pVITRO1-Trastuzumab-IgG1/κ was a gift from Andrew Beavil (Addgene plasmid # 61883; http://n2t.net/addgene:61883; RRID:Addgene_61883)[71], pDGM6 was a gift from David Russell (Addgene plasmid # 110660; http://n2t.net/addgene:110660; RRID:Addgene_110660)[72], Lenti-luciferase-P2A-Neo was a gift from Christopher Vakoc (Addgene plasmid # 105621; http://n2t.net/addgene:105621; RRID:Addgene_105621)[73], pMD2.G (Addgene plasmid # 12259; http://n2t.net/addgene:12259; RRID:Addgene_12259) and psPAX2 (Addgene plasmid # 12260; http://n2t.net/addgene:12260; RRID:Addgene_12260) were gifts from Didier Trono. mCAR-Vb2 lentiviral plasmid was generated via BbsI digestion of pSLCAR-CD19-BBz, followed by NEBuilder HiFi DNA Assembly with gBlock (IDT) containing scFv targeting TRBV20-1. To further stabilize scFv structure, amino acids at Kabat position VH44 and VL100 were changed to Cysteine. TCR-Vβ2 lentiviral plasmid was generated from pAW13.lentiguide.mcherry. In brief, the U6-gRNA scaffold fragment was cut out by KlfI/BsmBI double digestion, end repaired by Quick Blunting kit (NEB, E1201S), and self-ligated by T4 ligase (NEB, M0202S). Then the plasmid was double digested by BsiWI/MscI, followed by NEBuilder HiFi DNA Assembly (NEB, E2621S) with the TCR-Vβ2 PCR product from T cells of a Vβ2+ patient and gBlock (IDT) containing TCR-Vβ2 3' overlapping + P2A + mcherry 5' overlapping. Humanized CAR-Vβ2 (hCAR-Vβ2) plasmids were generated via BbsI digestion of pSLCAR-CD19-BBz, followed by NEBuilder HiFi DNA Assembly with humanized VH and VL gBlocks (IDT) containing VH44 and VL100 Cysteine replacement. pAAV.CMV.PI.EGFP.WPRE.bGH plasmid was used as AAV transfer vector backbone to generate CAR-expressing vector. Left and right homology arms of human *TRAC* region and CAR-GFP fragment were generated by PCR using gDNA from T cells of a healthy donor and a CAR-containing lentiviral plasmid as templates, respectively. WPRE structure from the

AAV backbone was removed, followed by sequential insertion of left homology arm, T2A, CAR, P2A, GFP, and right homology arm to generate CAR-expressing vector as pAAV.pTRA.T2A.CAR.P2A.GFP.bGH. pVITRO1-Trastuzumab-IgG1/k was used as the backbone to generate ADCC enhanced humanized anti-Vβ2 IgG1 antibody. pVITRO1-Trastuzumab-IgG1/k plasmid was digested by BsrGI/BspEI/ AvrII/NotI mixture, followed by gel purification of two digested fragments (2726 bp and 3761 bp). gblocks of the entire humanized anti-Vβ2 heavy chain with ADCC enhancing mutations (H268F/S324T/S239D/I332E)[56] and light chain variable region were synthesized by IDT and assembled with digested fragments above to form the humanized anti-Vβ2 IgG1 expressing plasmid using NEBuilder HiFi DNA Assembly.

## Human cell isolation and culture

Fresh peripheral blood was 1:1 diluted in room temperature (RT) 1xPBS and layered on Ficoll-Paque Premium (Cytiva, 17544202) in SepMate (Stem Cell Technologies, 85450) centrifuge tubes. The buffy coat was harvested according to SepMate instructions. After extensive PBS washing, PBMC were counted and then either directly cryopreserved in liquid nitrogen or immediately used for NK cell, pan T cell, CD4 T cell, CD8 T cell, or CTCL cell isolation by untouched MACS negative selection kits (Miltenyi, 130092657, 130096535, 130096533, 130096495). For CTCL and PTCL cell isolation, anti-CD26-biotin and/or anti-CD7-biotin (invitrogen, 13007982, BMS143BT) were added to remove CD26+ and/ or CD7+ cells based on the clinical phenotypes of the malignant CTCL cells in each patient. Isolated T cell subtypes were either directly cryopreserved in liquid nitrogen or cultured in T cell medium (RPMI, Gibco, 11875093) supplied with 10% heat-inactivated FBS (Gibco, 10438026), 20 ng/ml rhIL2, 5 ng/ml rhIL7 and 10 ng/ml rhIL15 (cytokines from R&D Systems, 202IL, 207IL,247ILB). The purity of isolated cell populations was assessed by flow cytometry and ranged from 95% to 98%.

## Antibody mass spectrometry (MS)

Antibody targeting human TCR Vβ2 (clone MPB2D5) was purchased from Beckman Coulter. High resolution MS was performed by Rapid Novor. Amino acid sequences of the antibody generated by MS were annotated to identify framework and CDR regions of variable domains of heavy and light chains.

## Lentivirus production

293 T cells were seeded overnight and then transfected by lentiviral vector, pMD2.G and psPAX2 in lipofectamine2000. 6–8 h post plasmid addition, fresh medium was added. Twenty-four hours later, the first supernatant was collected and stored at 4 °C. Fresh medium was added. 24 h later, the second supernatant was collected and combined with first one, followed by 0.4 µm filtering. Aliquots are held at −80 °C for long-term storage.

## Lentivirus transduction

Jurkat cells were re-suspended in lentiviral supernatant with 8µg/ml polybrene. Spinfection was performed 90 min × 1200 g at 32 °C, followed by changing to fresh medium. Primary human T cells freshly isolated or overnight recovered in cytokine-medium after thawing were stimulated by anti-CD3/CD28 beads at 1:1 ratio for 2 days. Then beads were magnetically removed and activated T cells were re-suspended in lentiviral supernatant containing cytokines and 8µg/ml polybrene. Spinfection was performed 120 min × 2000g at 32 °C, followed by changing to fresh medium containing cytokines. After overnight culture, T cells were centrifuged and re-suspended in fresh medium containing cytokines for further expansion.

## Gene knockout (KO)

sgRNAs targeting human TRAC, B2M, CIITA, TRBV12-3 and Fut8 were purchased from IDT. Culture medium was pre-warmed in a 37 °C incubator. Ribonucleoprotein (RNP) complex composed of Cas9

protein and sgRNA was formed at RT for 20 min. The supernatant was aspirated completely from the cell pellet which was then re-suspended in 20 µl P3 buffer for primary T cells, SE buffer for Jurkat cell line or SF buffer for ExpiCHO cell line (Lonza). Cell suspension, RNP complex and enhancer (IDT) were mixed well and transferred into 16-well strip tubes (Lonza). Electroporation was performed in Lonza 4D-x unit with program EO-115 for primary T cells, CL120 for Jurkat cell line or DU158 for ExpiCHO cell line. 100ul pre-warmed culture medium was immediately added to the cells after electroporation, and then transferred completely into pre-warmed medium for culture. sgRNAs are listed in Supplementary Table 2.

## CD3+ T cell depletion

Two days post TRAC KO, remaining CD3+ cells were removed using MACS anti-CD3 microbeads and LD columns (Miltenyi, 130050101) following the manufacturer's protocol. TRAC KO CD3- T cells were collected from the column flow through. Cell purity was assessed by flow cytometry and ranged from 95% to 98%.

## Sorting

Sterile sorting of target cells was performed on either an S3e cell sorter (Bio-Rad) or BD FACSAria II, with purity verification after each sorting. Cell purity ranged from 95% to 98%.

## CAR-T cell generation

Isolated CD8 T cells or pan-T cells were activated by anti-CD3/CD28 beads (Thermofisher, 11131D) for one or 2 days. Then the beads were removed, followed by gene (TRAC, B2M or CIITA) KO. After a 1–2-h rest in culture medium, CAR-containing lentivirus transduction was then performed. Transduced CAR-T cells were further expanded and differentiated in cytokine containing medium for 7–10 days before use in killing studies. Remaining CD3+ T cells were depleted 2 days post-TRAC KO. Purity was assessed by flow cytometry and ranged from 95% to 98%. Fresh cytokine-containing medium was added every 2–3 days.

## In vitro killing assay by CAR-T cells

Primary target cells isolated from patients or healthy donors were stained with CellTrace Violet dye (Thermofisher, C34557) following the manufacturer's instruction. $1 \times 10^4$ stained primary target cells, Jurkat-TRBV20-1 cells or Jurkat-TRBV6-2 cells were then seeded in 96-well U bottom plates. Effector CAR-T cells were mixed with the seeded target cells at effector to target (E:T) ratios of 0, 1:1 and 5:1. After overnight culture, cells were re-suspended in 1xPBS containing 2% FBS and 10µl CountBright Absolute Counting Beads (Thermofisher, C34950) for flow cytometry.

## In vitro CAR-Vβ2 T cell stimulation

CAR-Vβ2 T cells were co-cultured with CTCL cells from different patients or with Jurkat-TRBV20-1 cells for 6 h with or without eBioscience Protein Transporter Inhibitor Cocktail (Thermofisher, 00498093), then surface and/or intracellular cytokine staining was performed for flow cytometry.

## Short-term in vivo killing assay

On day 0 (D0), NSG mice were inoculated i.v. with $3 \times 10^6$ Jurkat-TRBV20-1 cells or $5–10 \times 10^6$ total CD4 T cells from Vβ2+ patients. On day 1 (D1) or day 3 (D3), $8–10 \times 10^6$ CAR-T cells per mouse were adoptively transferred i.v. into defined treatment groups. On day 4 (D4) or day10 (D10), mice were sacrificed and dissected to quantify remaining T lymphoma cells and CAR-T cells in spleen, bone marrow (BM) and blood by flow cytometry.

## Mouse dissection and cell isolation

Mice were euthanized by $CO_2$ inhalation. Blood was collected by cardiopuncture into EDTA-coated collection tubes. RBC lysis buffer

(Biolegend, 420301) was added for 10 min at RT. Spleens were smashed through a 70 μm strainer, followed by RBC lysis buffer incubation for 5 min at RT. Femur and tibia were isolated, the ends of each bone were removed and the bone marrow was flushed out using a syringe containing 1xPBS plus 2% FBS, and then smashed through a 70 μm strainer, followed by RBC lysis buffer incubation for 5 min at RT. Isolated cells were re-suspended in 1xPBS containing 2% FBS and kept on ice for further use.

## Flow cytometry

Cells were re-suspended in 100 μl 1xPBS containing FcR block (Biolegend, 422302) and Aqua Live/Dead fixable dye (Thermofisher, L34957) for 10 min at RT. Without wash, 100 μl antibody mixture in 1xPBS was added for 30 min at 4 °C. After washing in 1xPBS containing 2% FBS, cells were re-suspended in 200μl 1xPBS containing 2% FBS and 10 μl counting beads for flow cytometry detection were added. For intracellular staining, cells were fixed in 2% PFA for 20 min at RT and incubated with antibody mixture in 1xPermeablization buffer (Thermofisher, 00833356) for 30 min at RT. Antibodies used are as follows: Beckman Coulter: anti-Vβ-PE/FITC kit (Beckman, IM3497); Biolegend: anti-CD3-APC (300312), anti-CD4-APC-Cy7 (300518), anti-CD8-PE-Cy7 (344750), anti-CD45-PerCP-Cy5.5 (368504), anti-HLA-A/B/C-Pacific Blue (311418), anti-HLA-DR/DP/DQ-PE (361716), anti-4-1BB-BV421 (309819), anti-CD25-PE-Cy7 (302612), anti-CD69-PerCP-Cy5.5 (310926), anti-CD45RA-AF700 (304119), anti-CD45RO-BV510 (304246), anti-GZMB-APC (372204), anti-IFNγ-PE (506507), and anti-TNFα-APC-Cy7 (502944); Thermofisher: anti-human IgM-AF647 (314536) and anti-human IgG Fc-PE (MA110377). Protein L-PE (Cell Signaling Technology, #58036) and Lens Culinaris Agglutinin (LCA)-FITC (Thermofisher, L32475) staining were performed as surface antibody staining in 1xPBS. The antibody concentration applied in all assays adhere to the manufacturer's instructions and recommendations. The gating strategy used is presented in Supplementary Fig. 8.

## In silico humanization design

The BioPhi in silico modeling algorithm or a third-party contractor (mAbvice) were used for humanization design. In BioPhi, humanization settings were selected with Chothia Numbering and CDR definition and Humanization method 'Sapiens' or 'CDR Grafting'. The OASis prevalence threshold chosen was either relaxed (over 10%) or strict (over 90%). In each humanization algorithm, 4 humanized VH and 4 humanized VL chains were generated.

## Humanized CAR-T cell efficacy screening

Under each in silico humanization algorithm, 16 different humanized scFv fragments were generated and cloned into CAR-containing lentivirus plasmids. Pan-T cells isolated from a healthy donor were activated by anti-CD3/CD28 beads, followed by *TRAC* KO. Two days later, the remaining CD3+ cells were depleted by MACS. After a 2-h rest in culture, humanized CAR-Vβ2, CAR-CD19 or original mCAR-Vβ2 lentivirus transduction was performed. After a 7 day differentiation and expansion of CAR-T cells, in vitro screening for Vβ2+ cell killing efficiency was conducted using Vβ2+ CTCL cells from one Vβ2+ patient.

## CRISPR/AAV system to generate allogeneic CAR-T cells

To produce AAV, the AAV transfer plasmid and pDGM6 were co-transfected into 293 T cells that were harvested 3 days later. AAV in transfected 293 T cells were extracted and purified using the AAVpro purification kit (TaKaRa, Cat. 6675). Virus titer was then measured by qPCR. To generate CAR-expressing T cells using the AAV vector, T cells isolated from health donor were activated by anti-CD3/CD28 beads at 1:1 ratio for 2 days. Then beads were removed, followed by RNP-dependent *TRAC/B2M/CIITA* triple KO. After 1 h rest in complete T cell culture medium, T cells were resuspended in serum-free T cell culture medium containing cytokines and AAV (MOI = $2 \times 10^4$) at $1 \times 10^7$ cells/ml and incubated at 37 °C for 30 min. Then complete T cell culture medium containing 10% FBS, cytokines and homology recombination enhancer cocktail (M3814/TSA/XL413)[74] was added to a cell concentration of $1 \times 10^6$/ml. Homology recombination enhancer cocktail was removed after overnight or 1 day culture by replacing with fresh complete T cell medium containing cytokines. Remaining CD3+ cells were removed as above 2–3 days post KO.

## Immune reactivity assay

Human serum was collected from three healthy control (HC) donors (HC1, HC2 and HC3), one Vβ1+ CTCL patient, one Vβ2+ CTCL patient and one Vβ13.2+ CTCL patient. CAR-T cells were re-suspended in 50μl of 10-fold diluted serum and incubated at 4 °C for 30 min, followed by staining with anti-human IgM-APC secondary antibody at 4 °C for 20 min. GFP and IgM-APC were detected by flow cytometry.

## Mixed lymphocyte reaction (MLR) assay

*TRAC*-KO or triple-KO hCAR-Vβ2 T cells were generated from two healthy donors. CD8+ T cells were isolated from two additional donors. GFP + CAR-T cells and violet dye-labeled CD8+ T cells from different donors were mixed at different ratios overnight in culture, and then the number of remaining live CAR-T cells was quantified by flow cytometry using absolute counting beads.

## Long-term in vivo imaging of Jurkat-TRBV20-1-lucifer cells in NSG mice

$3 \times 10^6$ Jurkat cells engineered to express TRBV20-1 (Vβ2) and luciferase (Jurkat-TRBV20-1-lucifer) were i.v. inoculated into NSG mice. Three days later, $1 \times 10^7$ lenti-CAR-CD19, lenti-hCAR-Vβ2 or AAV-hCAR-Vβ2 T cells were i.v. transferred into the mice. Tumor burden was assessed once per week by IVIS (Lumina X5, PerkinElmer) bioluminescence imaging, 10 min after receiving 150 μl D-luciferin (PerkinElmer, 770504)/mouse (i.p.). Luciferin signals from both ventral and dorsal positions were quantified by ROI measurement (Living Image Software v4.8.0). Mouse body weight was monitored once per week initially and twice per week when tumors were evident.

## Cytokine optimization for hCAR-Vβ2 T cell expansion in vitro

One condition without any cytokines and 20 conditions with different cytokine combinations (Supplementary Table 1) were tested. The final concentration of each cytokine was as follows: IL2 (5 ng/ml, R&D), IL7 (5 ng/ml, R&D), IL12 (10 ng/ml, R&D), IL15 (10 ng/ml, R&D), IL18 (6.25 ng/ml, R&D) and IL21 (20 ng/ml, R&D). AAV transduced triple KO hCAR-Vβ2 T cells from a healthy donor were equally divided among the 21 conditions in triplicate. Live cells and viability were counted by trypan blue staining (Countess, Thermofisher) and cells were subcultured in fresh cytokine medium at $1 \times 10^5$ cells/200μl every 2–3 days. At day 7 of cytokine expansion, an aliquot of hCAR-Vβ2 T cells was used for flow cytometry detection of surface phenotype and intracellular cytokine production following a 6-h Jurkat-TRBV20-1 cell stimulation (E:T = 1:1). In addition, separate aliquots of $1 \times 10^4$ hCAR-Vβ2 T cells were repeatedly stimulated by $2 \times 10^4$, $5 \times 10^4$ and $1 \times 10^5$ Jurkat-TRBV20-1 cells at day 0, 1 and 2 of co-culture, respectively. Co-cultured cells were collected for flow cytometry at day 1, 2, 3 and 4 of co-culture to evaluate the number of live Jurkat-TRBV20-1 cells and hCAR-Vβ2 cells, and surface phenotype changes of hCAR-Vβ2 cells at the end of 4 days co-culture. At day 12 of cytokine expansion, hCAR-Vβ2 T cells were used for flow cytometry detection of surface phenotype and intracellular cytokine production at resting status without CAR engagement.

## Chronic stimulation and repeated killing assay

For each chronic stimulation and repeated killing cycle, hCAR-Vβ2 T cells generated from different cytokine expansion conditions were co-cultured with target Jurkat-TRBV20-1 cells at 1:2 ratio for

3 days. Then the remaining hCAR-Vβ2 T cells were purified through CD3+ Jurkat-TRBV20-1 cell magnetic depletion and an overnight killing assay was set up by mixing stimulated hCAR-Vβ2 T cells and fresh Jurkat-TRBV20-1 cells at a 1:1 ratio.

## Humanized anti-Vβ2 IgG1 production and purification

The ExpiCHO expression system kit (Thermofisher, A29129) was used to produce antibody following the manufacturer's instructions. Fut8 KO was performed in ExpiCHO cells by CRISPR-Cas9, followed by LCA staining and LCA- cell sorting to establish the Fut8 KO ExpiCHO cell line for production of antibody with enhanced ADCC effects[57]. Antibodies were purified by Magne Protein G beads (Promega) following the manufacturer's instruction. Eluted antibodies were dialyzed against PBS, followed by 0.22 μm sterilization filtering and storage at 4 °C. The concentration of purified antibodies was quantified by nanodrop with mass extinction coefficient of 13.7 at 280 nm for a 1% (10 mg/ml) IgG solution. Antibody purity was evaluated by 10% SDS-PAGE.

## Antibody competition assay

Vβ2+ CTCL cells from one patient were stained with a mixture of 1.25 μg/ml mouse anti-Vβ2-FITC and varying concentrations of purified humanized anti-Vβ2 IgG1, followed by anti-human IgG Fc-PE staining. The relative affinity of humanized anti-Vβ2 IgG1 to the original mouse anti-Vβ2 antibody was estimated by the concentration of humanized anti-Vβ2 IgG1 used to reach 50% of maximal mouse anti-Vβ2-FITC intensity.

## ADCC reporter bioassay

The Promega ADCC reporter bioassay kit was used to assess ADCC activity. In brief, $1 \times 10^4$ Vβ2+ CTCL cells from one patient were mixed with $1 \times 10^4$ effector Jurkat-NFAT-luciferase effector cells provided by the kit and varying concentrations of mouse anti-Vβ2 antibody or humanized anti-Vβ2 IgG1. After 6 h co-culture at 37 °C the assay reaction was quantified (Victor X-Light, PerkinElmer).

## In vitro ADCC of NK cells mediated by humanized anti-Vβ2 IgG1

PBMC isolated from two Vβ2+ CTCL patients were stained with Cell-Trace Violet dye (Thermofisher) following the manufacturer's instruction. $1 \times 10^4$ stained PBMC or Jurkat-TRBV20-1 cells were then seeded in 96-well U bottom plates. Purified effector NK cells from a healthy donor were mixed with the seeded target cells at effector to target (E:T) ratios of 0, 1:1, 5:1 or 10:1. Then 100 ng/ml mouse anti-Vβ2 antibody or humanized anti-Vβ2 IgG1 was added into cell mixture. After overnight culture, cells were re-suspended in 1xPBS containing 2% FBS and 10 μl CountBright Absolute Counting Beads (Thermofisher) for flow cytometry.

## Statistics

Prism 9 was used for statistical analysis, with two group comparisons by Student's t-test and three or more group comparisons using one-way or two-way ANOVA accordingly.

## Reporting summary

Further information on research design is available in the Nature Portfolio Reporting Summary linked to this article.

## Data availability

Data supporting the findings of the study are available within the Article, Supplementary Information and Source data file. Human-related data that were collected but not shown in the paper might be subject to confidentiality (e.g., sex and age). All outstanding data including hCAR-Vβ2 T and hAb-Vβ2 sequences are available upon request from the corresponding author, Michael Girardi (michael.girardi@yale.edu) due to the intellectual property protection applications that are currently under consideration for the disclosed innovations. Source data are provided with this paper.

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

## Acknowledgements

This work was supported by the R. S. Evans Foundation and the Cutaneous Lymphoma Foundation (received by MG). In addition, we extend our gratitude to the patients and healthy donors who generously donated their cells, enabling us to conduct this study.

## Author contributions

J.R., X.L. and M.G. designed experiments; J.R., X.L., J.M.L. and J.C. performed experiments; J.R., X.L., J.M.L., R.Q. and M.G. analyzed the data; K.R.C. coordinated patient-derived samples and IRB-approval; F.F. and M.G. screened patients and provided patient samples; J.R., X.L., J.M.L. and M.G. prepared figures and wrote the manuscript; all authors reviewed and edited the manuscript.

## Competing interests

J.R., X.L., J.M.L. and M.G are inventors on patent applications filed by Yale University. Application # 63/484,916. The remaining authors declare no competing interests.
