## [Peer Review File · Nature Communications]

Generation and optimization of off-the-shelf immunotherapeutics targeting TCR-V β 2+ T cell malignancyEditorial Note: Parts of this Peer Review File have been redacted as indicated to remove third-party material where no permission to publish could be obtained.

REVIEWER COMMENTS

Reviewer #1 (Remarks to the Author): with expertise in CAR-T cells

This manuscript describes the preclinical development of both CAR-based and ADCC-enhanced soluble therapeutics for T cell malignancies, a major unmet need. Specifically, the authors target Vbeta2, a dominant clonotype expressed on T cell cancers. The work is based on previously published antibody that the authors methodically optimize for CARs and soluble therapeutics that includes a humanization component. The data is, overall robust, and supports the conclusion, that the therapeutics are active. The results are presented in a logical manner. The methods are described in more than adequate detail. There are concerns with novelty as it relates to the broader field. There is novelty in the use of CTCL patient T cells and tumors which identified challenges with manufacturing of functional CARs from T cells derived from T cell cancer patients.

Comments:

- 1) The authors demonstrate clearance of the Vbeta2 expressing T cells during manufacturing. While this is indicative of a functional CAR, others have shown that this can result in a hypofunctional CAR product. Exploration of this could impact selection of healthy donors or suggest depletion of this clonotype prior to manufacturing as a means to improve potency.
- 2) The authors explore many different cytokine combinations during the manufacturing and arrive at specific combinations that they deem optimal. Expansion is only one (and likely not the most important) parameter that defines functions. The short-term cytotoxicity experiments can also be misleading. Thus, these results should be interpreted with caution. Other assays (repetitive stimulations, etc) would enhance these conclusions. Figure 5j suggests some of this was done or will be done?

Reviewer #2 (Remarks to the Author): with expertise in allogeneic CAR-T, genome editing

The paper investigates the production and modelling of CAR T cells against Vb2+ T cell

malignancy in a universal format, with CRISPR mediated disruption of TCR and MHC class I and II. Similar approaches against T cell cancers have targeted other T cell surface molecules, including the TCR via TRBC. There are already some programs in clinical phase testing, although perhaps not specifically against Vb2. The paper presents as an assembly of pickings from a variety of previously reported technical advances from across immunotherapy in recent years. These range from humanization of antibody fragments, to site-specific insertion of a CAR template into the TRAC locus, and multiplexed CRISPR/Cas9 editing to overcome HLA barriers. In addition to a universal T cell approach, there are also NK remedies proposed. While the work is technically sound, the question for publication in Nat Comms is whether this mixed bag of incremental developments delivers novelty or impact sufficient for the journal. Specific points are as follows:

1. Vb2 was found to be the most frequently expressed clone. Is there information on approximately on absolute patient numbers? ie. How many patients express Vb2 on all their malignant cells and how many might be suitable each year for a Vb2 CAR therapy year?
2. An initial section on generation of autologous Vb2 CAR T cells in CTCL seems to have been on a single patient. Should be reproduced in more subjects to draw conclusions
3. The generation and testing of triple-ko CAR T cells includes flow data but should include molecular confirmation of on target edits, and additional investigations for chromosomal changes (eg translocations as might be predicted) as well as screening for off target effects. Unless that data is already published for the same guide combinations.
4. Is any there any functional data confirming the immunological stealth of edited cells (eg MLRs) in vitro. The in vivo testing in NSG mice confirms potency against targets, but rejection experiments were not undertaken. They may be too challenging but perhaps could be discussed and literature cited.
5. The broad statement about AAV safety v lentiviral safety should be justified, given there has been no LV mediated leukemic transformation reported in T cells.

6. The section on ADCC mediated by NK cells seems is an interesting avenue, but seems out of place with the rest of the paper centred on CAR-T

7. Aspects relating to humanisation of the clone, and comparisons of culture conditions etc might all be considered background work to be shared in the supplemental.

Reviewer #3 (Remarks to the Author): with expertise in CAR-T, T cell malignancies

Investigators' manuscript titled "Generation and optimization of off-the-shelf immunotherapeutic targeting TCR-Vb2+ T cell malignancy" described the approach of rapid generation of an off-the-shelf allogeneic chimeric antigen receptor (CAR)-T platform through CRISPR-induced triple knockout (TRAC, B2M, and CIITA) to eliminate T cell-dependent graft-versus-host (GVH) and host-versus-graft (HVG) reactivity, targeting the clone-specific TCR Vb chain malignant T cell killing while limiting normal cell destruction.

This approach has an obvious advantage over the current targeting CD4 base CAR T for T cell malignancies. Their CAR T will have CD4 and CD8 phenotypes but can avoid CAR T fratricide and normal T cell depletion. It fills the unmet safety gap in the field of immunotherapies.

The manuscript's pre-clinical strategies are well-designed and convincing. The conclusion is precise, and the reviewer has no major concerns.

Minor concerns:

Investigators should, at least, discuss the potential risk of malignant transformation of their CRISPR-edited CAR.

Suggest work: Deep sequencing 5-10 separately transduced CAR products that may suggest some level of safety confidence regarding the transformation risk.

Review recommendation: Accept this MS with minor revisions.

Reviewer #4 (Remarks to the Author): with expertise in CAR-T, T cell malignancies

The report is well written and comprehensive attempt target a small subset of mature T cell lymphomas via CART to VB2 which is only expressed in 3-9% of normal T cells and ~5-15% of malignant T cells (20-25% of CTCL; Fig 1A). Thus, this approach and optimized reagent (humanized and modified/optimized CAR-VB2. Thus, although the authors demonstrate high specificity and efficacy of h/mCART-VB2 for VB2+ T cell malignancies in vitro and in vivo (NSG mice) the impact on its translation to the clinic is questionable since other immunotherapies targeting mature T cell malignancies such as CART-TCB1 (from Autolus; Auto4). Furthermore the paper is somewhat disjointed with a portion related to in vitro and limited in vivo effects of mCARTVB2 and hCARTVB2 on very limited CTCL PDX and Jurkats expressing TCR with VB2, humanization of the CAR, generating triple KO (TRAC, B2, CIITA) CART, AAV KI CRISPR experiments, the effect infinite combinations of cytokines on expansion, IFN generation, co-stim molecule expression and CD45RA/RO expression and finally, the use of an antibody with ADCC properties to enhance NK mediated killing of the same target population. The linkage of many of these to a common theme are not well established. In fact the cytokine experiments are difficult to interpret and the NK Ab studies (Fig 6) seem unrelated to the major focus of the paper and even redundant.

Minor:

1. CTCL is very rare and for many patients treatments, when necessary are well tolerated and associated with long progression free survivals and good quality of life. For the more common PTCLs, the incidence of VB2+ non-CTCL PTCLs is very low. How do the authors see this complicated therapy fit in with many other therapies and Autolus CARTs in play?
2. The authors state that eliminating 50% of the T cell population with the Autolus approach would result in significant problems and immunodeficiency. What evidence that this would be an issue since the number of almost infinite TCR diversity would only be reduced by 50% which would likely not impact immune responses.
3. Killing curves shown in Figure 1h are very unimpressive. What is the reason for this? Could the CARTs be exhausted during manufacturing due to constant exposure of target (VB2+ T cells) during expansion?
4. A major tenant of this paper is the need to generate "off-the shelf" TRAC KO CART since

apparently endogenous (autologous T cells) from CTCL patients do not perform as well as normal allogeneic T cells or TRAC KO allogeneic T cells (results of Fig 1c/e vs Fig 1d). Is this conclusion made from a single patient and a single manufacturing generation? Is this the case for other PTCL patients? The only setting where this has been suggested is in CLL where even there the data is not definitive and only suggestive in spite of many samples and donors being analyzed. The need to move to off the shelf is justified by this but this is not demonstrated clearly since only a single patient is studied. The authors could do additional well controlled studies trying to confirm their preliminary observation and also no mechanistic studies are done to determine why this might be.

5. Confirmatory experiments demonstrating the defect in autologous CTCL or PTCL CART compared to healthy or to allogeneic triple KO CART could also be tested with other CARs such as CAR19 and targets CD19+ B cell lines (Ramos etc) vs VB2 and VB2+ targets.

6. Triple KO experiments and humanization studies done in Figures 2 and 3 are well organized and the data is compelling. The issue is related to the lack of GvHD suggested by the authors. In Figure 4 there is no observation of weight loss of formal mouse GvHD staging. One would expect that the NC and lentiviral CARTVB2 to have significant weight loss and death since both have intact TCR. It is clear that the lentiviral CARTVB2 effectively clear tumor like the AAV KI and TRAC KO model but both lentiviral and AAV groups have similar anti-tumor effects (expected) but also the same survival (unexpected). So why is this and what are the GvHD scores for these groups of NSG mice? If lentiviral CARTVB2 do not induce any xenogeneic GvHD then this would be very interesting. Also the followup for these experiments is very limited. What is the survival in the lentiviral and AAV groups after day +37. Longer followup would be essential to look at relapse rates (if relapsed tumors are antigen + or -) and GvHD effects.

7. The data shown in Extended Figure 2 demonstrate high efficiency of TRAC deletion (>90%), and B2/Class I deletion (>95%) but low efficiency of Class II deletion (CIITA). Would be important using NHEJ sequencing what the actual deletion of CIITA is...may take some time for Class II to decrease at the protein level or simply that the editing is poor. Also, no off target analyses are presented in these cells subjected to multiplex editing with both HR and NHEJ approaches.

8. The cytokine data presented in part in Figure 5 is difficult to interpret. Although the exercise is admirable there are no strong conclusions that can be made unless this were

done many times with many different donors then the general conclusions summarized or proposed in Figure 5J. Also there is really limited meaning in general without any in vivo validation studies. For instance one or two validation runs testing the best and worse cytokine combinations on in vivo killing would be really provide important context for all of these comprehensive in vitro studies.

9. The data presented in Figure 6 is incremental at best and not well related to the rest of the focus of this manuscript and provides a distraction and not any unifying hypothesis or important mechanistic studies. Seems like a completely different topic.

Revision of manuscript “**Generation and optimization of off-the-shelf immunotherapeutics targeting TCR-V β 2+ T cell malignancy**” (NCOMMS-23-10140-T) by Ren et al.

Dear Reviewers:

We sincerely appreciate the reviewers’ thorough reviews and insightful comments and suggestions. We have addressed each and provide here a point-by-point response below, including delineation of the corresponding specific changes made to the manuscript. We are pleased to provide these responses, including revised figures, that we believe have all led to a substantially improved manuscript.

Reviewer Comments:

Reviewer #1 (Remarks to the Author): *with expertise in CAR-T cells*

Rev. 1, Point 1. The authors demonstrate clearance of the Vbeta2 expressing T cells during manufacturing. While this is indicative of a functional CAR, others have shown that this can result in a hypofunctional CAR product. Exploration of this could impact selection of healthy donors or suggest depletion of this clonotype prior to manufacturing as a means to improve potency.

We appreciate the reviewer’s questioning a potential CAR-T manufacturing issue, but believe there may have been some misunderstanding of our methods and data. What we have demonstrated in vitro (*after* CAR-T generation, Fig. 1d-f) is the specific killing of V β 2+ T cells while leaving all other V β + T cells unaffected. These data show that the generated V β 2-targeting CAR-T are exquisitely specific in distinguishing across other TCR-V β s. In our original results section entitled “Allogeneic TRAC/B2M/CIITA triple-knockout mCAR-V β 2 T cells show specific V β 2 targeting while minimizing GVH effects”, we clarified the rationale for this approach: “Towards the development of off-the-shelf allogeneic mCAR-V β 2 T cells from healthy donors, we first knocked out the TRAC gene of purified CD8 T cells with over 70% efficiency, followed by residual CD3+ cell depletion (Extended Data Fig. 2a).” Please note that this occurs even before the expression of the CAR construct on the T cells. Thus, there is no concern that the generated CAR-V β 2 T cells will express V β 2, or become hypofunctional due to activation and elimination of V β 2+ cells during generation. For further clarification on our approach, we have amended the manuscript by including the following text:

Added to the Results section: Our triple-KO approach also reduces the concern that the generated CAR-V β 2 T cells will express V β 2, or become hypofunctional due to activation and elimination of V β 2+ cells during CAR generation.

In addition, it is possible that the reviewer may also have been considering the effects of repeated exposure of the generated CAR-T to the V β 2 target (i.e. repeated stimulation that might lead to decreased efficacy). We have now demonstrated the persistence of killing by the CAR-T (see immediately below in **Point 2**).

Rev. 1, Point 2. The authors explore many different cytokine combinations during the manufacturing and arrive at specific combinations that they deem optimal. Expansion is only one (and likely not the most important) parameter that defines functions. The short-term cytotoxicity experiments can also be misleading. Thus, these results should be interpreted with caution. Other assays (repetitive stimulations, etc) would enhance these conclusions. Figure 5j suggests some of this was done or will be done?

We appreciate and agree with the reviewer’s insightful and informed comments. In response, we have now assessed the degree to which chronic stimulation via short-term repeated

exposure might adversely affect CAR-V β 2 function. In Fig. 5c, we showed how IL12-conditioned CAR-V β 2 T cells sustain short-term repeated killing efficacy. In new data (Fig. 5b, f) using CAR-V β 2 T cells generated from two additional donors under seven different cytokine expansion conditions, we further chronically stimulated CAR-V β 2 T cells with Jurkat-TRBV20-1 (V β 2+) cells every three days, assessing their cytotoxic (killing) capacity after each cycle. These data show that CAR-V β 2 T cells generated from both new donors using IL7+IL15+IL12 conditioning consistently also supported CAR-T expansion and sustained Jurkat-TRBV20-1 cell killing ability (along with other cytokine combinations). We have now included this additional data (new Fig. 5f panel, included below), have tempered some of the language as supportive conditions (as opposed to “optimized”), and have amended the manuscript by completely revising the results section now entitled “Cytokine combinations for allogeneic hCAR-V β 2 T cell expansion and differentiation.”

New Fig. 5f panel:

Reviewer #2 (Remarks to the Author): with expertise in allogeneic CAR-T, genome editing

Rev. 2, Point 1. The paper investigates the production and modelling of CAR T cells against V β 2+ T cell malignancy in a universal format, with CRISPR mediated disruption of TCR and MHC class I and II. Similar approaches against T cell cancers have targeted other T cell surface molecules, including the TCR via TRBC. There are already some programs in clinical phase testing, although perhaps not specifically against V β 2.

We appreciate the reviewer's comparison of our approach of CAR-T targeting the specific TCRV β family (expressed on ~4-9% of normal T cells) to targeting TRBC1 and TRBC2 (each expressed on ~50% of normal T cells), and we had in fact cited that work while explaining the major potential advantages of our strategy targeting TCRV β : increased specificity with much less depletion of normal T cells. Even though anti-TRBC is in clinical trials, this does preclude our targeting of a variable region of the TCR as a viable strategy with potential advantages in specificity and safety. Please note that currently (per clinicaltrials.gov) there are no clinical trials with CAR-T or therapeutic antibodies that target TCRV β 2, or any other TCR β variable chain. For these reasons, we believe our work is novel and not simply incremental over the anti-TRBC approach.

Rev. 2, Point 2. The paper presents as an assembly of pickings from a variety of previously reported technical advances from across immunotherapy in recent years. These range from humanization of antibody fragments, to site-specific insertion of a CAR template into the TRAC locus, and multiplexed CRISPR/Cas9 editing to overcome HLA barriers. In addition to a universal T cell approach, there are also NK remedies proposed. While the work is technically sound, the question for publication in Nat Comms is

whether this mixed bag of incremental developments delivers novelty or impact sufficient for the journal.

We agree that several of the critical advances to immunotherapeutics that we have utilized were originally developed by colleagues, and we have appropriately cited each. Please note that each of these were selected, applied, and studied in our system in combination to provide an optimized strategy that we believe would best position our anti-V β 2 CAR-T (and corresponding therapeutic antibody) to treat V β 2+ T cell malignancies, and support translation of these preclinical studies towards clinical trials. Specifically, each of the aforementioned techniques were chosen to address the treatment of T cell lymphomas (and leukemias) that present a particular set of hurdles (not present, for example, in the treatment of B cell malignancies with anti-CD19 agents): to overcome the T cell dysfunction that is prevalent across T cell malignancies¹⁻⁵, and to avoid transfection of malignant T cells that may occur if we utilized autologous T cells as the source for CAR-T⁶. Thus, we are studying in preclinical models the feasibility of anti-V β 2 CAR-T derived from allogeneic donors (utilizing TRAC and HLA editing to minimize off-target effects and CAR-T rejection), as opposed to utilizing patient-derived CAR-T cell generation. For similar reasoning, we have presented the feasibility of a corresponding anti-V β 2-specific (derived from the same antigen recognition sequence) therapeutic antibody that utilizes a strategy combining Fc ADCC enhancement plus allogeneic NK.

1. Torrealba, M.P., *et al.* Chronic activation profile of circulating CD8+ T cells in Sezary syndrome. *Oncotarget* 9, 3497-3506 (2018).
2. Stolarencu, V., *et al.* Cellular Interactions and Inflammation in the Pathogenesis of Cutaneous T-Cell Lymphoma. *Front Cell Dev Biol* 8, 851 (2020).
3. Yawalkar, N., *et al.* Profound loss of T-cell receptor repertoire complexity in cutaneous T-cell lymphoma. *Blood* 102, 4059-4066 (2003).
4. Tanaka, T., *et al.* Opportunistic Infections in Patients with HTLV-1 Infection. *Case Rep Hematol* 2015, 943867 (2015).
5. Moriyama, K., *et al.* Immunodeficiency in preclinical smoldering adult T-cell leukemia. *Jpn J Clin Oncol* 18, 363-369 (1988).
6. Kozani, P., *et al.* CAR-T cell therapy in T-cell malignancies: Is success a low-hanging fruit? *Stem Cell Research & Therapy* 12, 527 (2021).

(Note: references above are also included in the manuscript).

Rev. 2, Point 3. V β 2 was found to be the most frequently expressed clone. Is there information on approximately on absolute patient numbers? How many patients express V β 2 on all their malignant cells and how many might be suitable each year for a V β 2 CAR therapy year?

We appreciate the reviewer is asking what number of patients might be eligible for anti-V β 2 CAR-T, as this has implications for meeting unmet needs, clinical trial feasibility, and efficacy related to target expression. Please note that we have described in our manuscript discussion an overall strategy to develop a set of therapeutics that would target several more of the most common V β s that, if successful, would eventually provide for personalized treatments beyond those with V β 2+ T cell malignancies. We had also provided our own data on V β 2 usage in CTCL (~25% of 72 patients, as presented in our Fig. 1A), and none of these patients showed expansion beyond the one V β indicated. We have also highlighted from the literature relevant data on V β 2 usage in this and other T lymphomas and leukemias that indeed also shows preferential V β 2 usage:

In a recently published metanalysis of 574 T cell lymphoma patients⁷ [Iyer *et al.* *Blood Advances*, 2022] that states: "There was a strong preferential usage of certain V β and V α segments across different TCLs. The most striking was TRBV20-1" (V β 2) in a large proportion of mRNA expressed clonotypes in CTCL (MF and SS), ATCLL, other TCL, and PTCL.

[Figure redacted]

(Note: Adapted from Iyer et al. *Blood Advances*, 2022⁷ for reviewer response only.)

In another study that used flow cytometry, V β 2 usage was the most prevalent in established T cell cancer lines as well as identified in mature T cell lymphoma patients⁸. In a third study, flow cytometry detection of V β usage in T-cell large granular lymphocytic leukemia (T-LGLL) showed V β 2 was utilized in 23.5% of isolates⁹. Thus, multiple sources indicate that V β 2 is one of (if not the most) prevalent V β used across T cell malignancies justifying it as an appropriate first target for anti-V β family immunotherapeutics that might more specifically target the malignant T cells (and spare the vast majority of normal T cells) in appropriately matched patients. We might also consider that an expanding set of anti-V β therapeutics might one day allow for personalized therapy even for those rare patients that show multiple malignant T cell clones. Thus, while we believe that an accurate set of calculations of proportional eligibility for anti-V β 2 CAR-T therapy across all the subtypes of T cell leukemias and lymphomas is beyond the scope of this report, we nonetheless contend that there is far reaching potential beyond CTCL.

7. Iyer, A., *et al.* Clonotype pattern in T-cell lymphomas map the cell of origin to immature lymphoid precursors. *Blood Advances* 6, 2334-2345 (2022).

8. Langerak, A.W., *et al.* Molecular and flow cytometric analysis of the Vbeta repertoire for clonality assessment in mature TCRalpha beta T-cell proliferations. *Blood* 98, 165-173 (2001).

9. Hsieh, Y.C., *et al.* A comparative study of flow cytometric T cell receptor Vbeta repertoire and T cell receptor gene rearrangement in the diagnosis of large granular lymphocytic lymphoproliferation. *Int J Lab Hematol* 35, 501-509 (2013).

(Note: references above are included in the manuscript).

Rev. 2, Point 4. An initial section on generation of autologous V β 2 CAR T cells in CTCL seems to have been on a single patient. Should be reproduced in more subjects to draw conclusions.

To support our findings, we have now repeated the autologous CAR-V β 2 T cell killing assay using two additional patient donors, which also consistently showed compromised killing efficacy (below). These results are combined with our originally presented data and shown in a new Extended Data Fig. 1h.

Rev. 2, Point 5. The generation and testing of triple-ko CAR T cells includes flow data but should include molecular confirmation of on-target edits, and additional investigations for chromosomal changes (e.g. translocations as might be predicted) as well as screening for off-target effects. Unless that data is already published for the same guide combinations.

Yes, the sgRNA for TRAC was previously validated¹⁰ with no detectable off-target effects. For sgRNAs for B2M and CIITA, these were selected from the recommended top-ranking sequences pre-designed by IDT which factors in an off-target score. We agree with the reviewer that prior to any potential clinical application we will still need to optimize and comprehensively evaluate genetic toxicity/off-target effects of our triple-KO system, including using high-fidelity Cas9 protein and screening multiple sgRNAs for B2M and CIITA regarding on-target efficacy and off-target toxicity, and adjusting dose/electroporation parameters. Importantly, the currently used sgRNA sequences have also shown efficient target protein KO, and we now provide additional functional (MLR) validation (please see response **Rev. 2, Point 6**, immediately below). Given the reviewer's important point, we have added the following statement to the the Discussion section:

“The sgRNAs targeting TRAC, B2M and CIITA in this manuscript are only used for proof-of-principle, so potential off-target effects would need to be carefully screened prior to clinical product development. Optimization of such might include using high-fidelity Cas9 protein, screening multiple B2M and CIITA sgRNAs, and adjusting electroporation parameters.”

10. Eyquem, J., *et al.* Targeting a CAR to the TRAC locus with CRISPR/Cas9 enhances tumour rejection. *Nature* 543, 113-117 (2017).

Rev. 2, Point 6. Is any there any functional data confirming the immunological stealth of edited cells (eg MLRs) in vitro. The in vivo testing in NSG mice confirms potency against targets, but rejection experiments were not undertaken. They may be too challenging but perhaps could be discussed and literature cited.

We appreciate this important consideration raised by the reviewer, including regarding the challenges of in vivo experiments to address this point. We were nonetheless able to perform an in vitro MLR experiment combining TRAC-KO or triple-KO CAR-T cells from two healthy donors, with CD8+ T cells isolated from either a CTCL patient or a different healthy donor. These two MLR reactions with different donor pairs consistently showed that killing efficiencies of allogeneic triple-KO CAR-T cells by CD8+ T cells were significantly reduced relative to allogeneic TRAC-KO CAR-T cells (shown below, and as new Extended Data Fig. 4d), data consistent with our allogeneic triple-KO CAR-T being more stealthy (i.e. with less HVG reactivity) – although we agree that clinical trials would be necessary to confirm this in treated patients. This concept is especially relevant given that T cell malignancy patients show T cell

functional deficiencies and CAR-T protocols would likely require induction chemotherapy that would further compromise T cell-mediated immunity against the CAR-T.

To further address this point, we have added the following to the Results section: “To further assess the effectiveness of our triple-KO strategy for reducing immune reactivity, we utilized an in vitro mixed lymphocyte reaction (MLR) combining CD8+ T cells from multiple donors with allogeneic triple-KO hCAR-Vβ2 versus TRAC-KO hCAR-Vβ2 (Extended Data Fig. 4d).”

Rev. 2, Point 7. The broad statement about AAV safety v lentiviral safety should be justified, given there has been no LV mediated leukemic transformation reported in T cells.

While we agree with the reviewer that LV-mediated leukemic transformation may be less likely in mature T cells, as supported in a study using mouse derived cells¹¹, this remains a theoretical risk for LV-mediated engineering of human T cells. We have modified our wording in the manuscript as follows: The result title ‘CRISPR-AAV system for safer CAR-Vβ2 engineering’ has been changed to ‘CRISPR-AAV system as an alternative for CAR-Vβ2 engineering’. The line ‘AAV-induced precise CAR genome integration combined with CRISPR KO may be safer for clinical use’ has been changed to ‘AAV-induced precise CAR genome integration combined with CRISPR KO may mitigate potential safety concerns of the lentiviral system’. In the Discussion, we added the line ‘However, whether (and to what degree) lentiviral mediated integration increases the risk of leukemic transformation in human CAR-T or other T cells will require long-term follow-up clinical studies.’

11. Newrzela, S., *et al.* Resistance of mature T cells to oncogene transformation. *Blood* 112, 2278-2286 (2008).

Rev. 2, Point 8. The section on ADCC mediated by NK cells seems is an interesting avenue, but seems out of place with the rest of the paper centered on CAR-T

The high-level primary purpose of this manuscript is to provide readers a feasible, personalized, off-the-shelf cell therapy platform for T cell malignancies, using anti-Vβ2 therapeutics as a proof-of-principle for targeting other Vβs more generally. We also wanted to show that the antigen recognition sequence / structure of our anti-Vβ2 CAR-T (and any other anti-Vβ CAR we develop) may be rapidly adapted to a therapeutic antibody agent, as such a strategy offers the potential to treat patients with personalized agents matched to their Vβ expression, as well as to specific clinical settings, i.e. different TCL types and stages may be better suited for treatment with a CAR-T versus a therapeutic antibody agent. As is generally known, patients with T cell malignancies often show increased opportunistic infections due to compromises in normal T cell

and NK numbers and/or function¹²⁻²⁵. Specifically, in CTCL patients, defects in cell-mediated immunity result in a heightened susceptibility to recurrent bacterial (e.g. Staph) and viral infections (e.g. herpes zoster). Published research has highlighted a decrease in NK cell functionality in these patients, which potentially weakens the innate immune response against both malignant cells and pathogens. Thus, our two strategies are justified and complimentary. Donor (allogeneic) T cells with their more efficient transduction efficiency (relative to NK cell transduction) are suitable for CAR engineering directly, while donor NK cells can utilize infused therapeutic antibody via their Fc receptors as an alternative strategy. We have edited the Results sections so that readers can better appreciate these complimentary strategies. Specifically, we have added to the Results section under the heading “Humanized anti-V β 2 IgG1 antibody with enhanced ADCC as another therapeutic option for V β 2+ T cell malignancy”, the following text:

“To expand the potential for anti-V β therapeutics to treat patients with personalized agents matched not only to their V β expression, but as well to specific clinical settings (i.e. different TCL types and stages may be better suited for treatment with a therapeutic antibody versus a CAR-T), we also developed an allogeneic NK-cell based ADCC platform. As is generally known, patients with T cell malignancies often show increased opportunistic infections due to compromises in normal T cell and NK numbers and/or function. We reasoned that allogeneic T cells with their more efficient transduction efficiency (relative to NK cell transduction) are suitable for CAR engineering directly, while donor NK cells could utilize infused therapeutic antibody via their Fc receptors as an alternative and complimentary therapeutic strategy.”

12. Kawano, N., et al. Clinical features and treatment outcomes of opportunistic infections among human T-lymphotrophic virus type 1 (HTLV-1) carriers and patients with adult T-cell leukemia-lymphoma (ATL) at a single institution from 2006 to 2016. *J Clin Exp Hematop.* 2019 Dec; 59(4): 156–167.
13. King ALO, et al. Factors Associated With In-Hospital Mortality in Mycosis Fungoides Patients: A Multivariable Analysis. *Cureus.* 2022 Aug 15;14(8):e28043
14. Vonderheid EC, et al. Herpes zoster-varicella in cutaneous T-cell lymphomas. *Arch Dermatol.* 1980 Apr;116(4):408-12.
15. Talpur R, et al. Prevalence and treatment of *Staphylococcus aureus* colonization in patients with mycosis fungoides and Sézary syndrome. *Br J Dermatol.* 2008 Jul;159(1):105-12.
16. Blaizot R, et al. Infectious events and associated risk factors in mycosis fungoides/Sézary syndrome: a retrospective cohort study. *Br J Dermatol.* 2018 Dec;179(6):1322-1328.
17. David C. Rhew, et al. “Infections in Patients with Chronic Adult T-Cell Leukemia/Lymphoma: Case Report and Review.” *Clinical Infectious Diseases*, vol. 21, no. 4, 1995, pp. 1014–16.
18. Girardi M, et al. The pathogenesis of mycosis fungoides. *N Engl J Med.* 2004 May 6;350(19):1978-88.
19. Krejsgaard, T., et al. Regulatory T cells and immunodeficiency in mycosis fungoides and Sézary syndrome. *Leukemia* 26, 424–432 (2012).
20. Kim EJ, et al. Immunopathogenesis and therapy of cutaneous T cell lymphoma. *J Clin Invest.* 2005 Apr;115(4):798-812. doi: 10.1172/JCI24826. Erratum in: *J Clin Invest.* 2007 Mar;117(3):836.
21. Harkins CP, et al. Cutaneous T-Cell Lymphoma Skin Microbiome Is Characterized by Shifts in Certain Commensal Bacteria but not Viruses when Compared with Healthy Controls. *J Invest Dermatol.* 2021 Jun;141(6):1604-1608.
22. Laroche L, et al. Decreased natural-killer-cell activity in cutaneous T-cell lymphomas. *N Engl J Med.* 1983;308(2):101-102.
23. Wood NL, et al. Depressed lymphokine activated killer cell activity in mycosis fungoides. A possible marker for aggressive disease. *Arch Dermatol.* 1990;126(7):907-913.
24. Yoon JS, et al. IL-21 enhances antitumor responses without stimulating proliferation of malignant T cells of patients with Sézary syndrome. *J Invest Dermatol.* 2008;128(2):473-480.
25. Rook AH, et al. IL-12 reverses cytokine and immune abnormalities in Sezary syndrome. *A H Rook; J Immunol* (1995) 154 (3): 1491–1498.

Rev. 2, Point 9. Aspects relating to humanization of the clone, and comparisons of culture conditions etc might all be considered background work to be shared in the supplemental data.

To address the reviewer's suggestions, we have moved Fig. 3a and 3b to Extended Data Fig. 3a and 3b. For cytokine expansion conditions, we reasoned that optimization is a critical parameter for a successful and durable CAR-T treatment, so have maintained it in **Figure 5**. To strengthen our cytokine optimization findings, we further evaluated in vitro cell expansion and chronic/repeated killing abilities of CAR-T cells generated from two additional healthy donors. (Please note that these changes are detailed above in the response to **Rev. 1, Point 2**). These data show that CAR-V β 2 T cells generated from both new donors using IL7+IL15+IL12 conditioning consistently also supported CAR-T expansion and sustained Jurkat-TRBV20-1 cell killing ability (along with other cytokine combinations). We have now included this additional data (new Fig. 5f panel), have tempered some of the language as supportive conditions (as opposed to "optimized"), and have amended the manuscript by completely revising the results section now entitled "Cytokine combinations for allogeneic hCAR-V β 2 T cell expansion and differentiation." We defer to the editor regarding final decisions of placement of figures.

Reviewer #3 (Remarks to the Author): with expertise in CAR-T, T cell malignancies

Investigators' manuscript titled "Generation and optimization of off-the-shelf immunotherapeutic targeting TCR-V β 2+ T cell malignancy" described the approach of rapid generation of an off-the-shelf allogeneic chimeric antigen receptor (CAR)-T platform through CRISPR-induced triple knockout (TRAC, B2M, and CIITA) to eliminate T cell-dependent graft-versus-host (GVH) and host-versus-graft (HVG) reactivity, targeting the clone-specific TCR V β chain malignant T cell killing while limiting normal cell destruction.

This approach has an obvious advantage over the current targeting CD4 base CAR T for T cell malignancies. Their CAR T will have CD4 and CD8 phenotypes but can avoid CAR T fratricide and normal T cell depletion. It fills the unmet safety gap in the field of immunotherapies. The manuscript's pre-clinical strategies are well-designed and convincing. The conclusion is precise, and the reviewer has no major concerns. Review recommendation: Accept this MS with minor revisions.

We sincerely appreciate the reviewer's encouraging comments and recognition of the novelty and potential impact of our findings and manuscript.

Minor concerns:

Rev. 3, Point 1. Investigators should, at least, discuss the potential risk of malignant transformation of their CRISPR-edited CAR.

We appreciated the reviewer's suggestion regarding the potential risk of malignant transformation of CRISPR-edited CAR-T cells due to possible off-target effects. We have addressed these concerns above in response to **Rev. 2, Point 5** and **Rev. 2, Point 7**.

Reviewer #4 (Remarks to the Author): with expertise in CAR-T, T cell malignancies

Rev. 4, Point 1. The report is well written and comprehensive attempt target a small subset of mature T cell lymphomas via CART to VB2 which is only expressed in 3-9% of normal T cells and ~5-15% of malignant T cells (20-25% of CTCL; Fig 1A). Thus, this approach and optimized reagent (humanized and modified/optimized CAR-VB2). Thus, although the authors demonstrate high specificity and efficacy of h/mCART-VB2 for VB2+ T cell malignancies in vitro and in vivo (NSG mice) the impact on its translation to the clinic is questionable since other immunotherapies targeting mature T cell malignancies such as CART-TCB1 (from Autolus; Auto4).

Regarding comparison of our approach of CAR-T targeting the specific TCRV β family to targeting TRBC1 and TRBC2, please see response (and included references) addressing **Rev. 2, Point 1** (above).

Regarding frequency of V β 2 usage across T cell malignancies, please see response (and included references) addressing **Rev. 2, Point 3** (above).

Rev. 4, Point 2. Furthermore the paper is somewhat disjointed with a portion related to **in vitro** and **limited in vivo** effects of mCARTVB2 and hCARTVB2 on very limited CTCL PDX and Jurkats expressing TCR with VB2, humanization of the CAR, generating triple KO (TRAC, B2, CIITA) CART, AAV KI CRISPR experiments, the effect infinite combinations of cytokines on expansion, IFN generation, co-stim molecule expression and CD45RA/RO expression. The use of an antibody with ADCC properties to enhance NK mediated killing of the same target population. The linkage of many of these to a common theme are not well established. In fact, the cytokine experiments are difficult to interpret and the NK Ab studies (Fig 6) seem unrelated to the major focus of the paper and even redundant.

Regarding our inclusion rationale for each of our engineering strategies (and their combinations) included in this manuscript, please see the detailed response addressed in **Rev. 2, Point 2** (above).

Regarding our inclusion of both of our therapeutic strategies, e.g. anti-V β 2 CAR-T and therapeutic antibody (with NK cells), please see response addressed in **Rev. 2, Point 8** (above).

Regarding our inclusion of data regarding cytokine culture conditions, please note that changes are detailed above in the response to **Rev. 1, Point 2**. Our new data show that CAR-V β 2 T cells generated from both new donors using IL7+IL15+IL12 conditioning consistently also supported CAR-T expansion and sustained Jurkat-TRBV20-1 cell killing ability (along with other cytokine combinations). We have now included this additional data (new Fig. 5f panel), have tempered some of the language as supportive conditions (as opposed to “optimized”), and have amended the manuscript by completely revising the results section now entitled “Cytokine combinations for allogeneic hCAR-V β 2 T cell expansion and differentiation.” We defer to the editor regarding final decisions of placement of figures.

New Fig. 5f panel

Minor Points:

Rev. 4, Point 3. CTCL is very rare and for many patients’ treatments, when necessary are well tolerated and associated with long progression free survivals and good quality of life. For the more common PTCLs, the incidence of VB2+ non-CTCL PTCLs is very low.

How do the authors see this complicated therapy fit in with many other therapies and Autolus CARTs in play?

Regarding the novelty and scope of the application in this manuscript and the frequency of V β 2 usage in non-CTCL TCLs, please see the detailed responses addressed in **Rev. 2, Point 1 and 3** (above).

We appreciate the reviewer's comments that CTCL that is in many cases an indolent lymphoma and in the common form of (and early stages of) mycosis fungoides that many treatments exist. However, we disagree that for advanced stages and aggressive forms of CTCL [e.g. MF with tumor stage (T3), leukemic/erythrodermic Sézary syndrome (T4), primary cutaneous anaplastic large cell lymphoma (pcALCL), CD8+ aggressive cytotoxic T cell lymphoma, etc] that available approved treatments are generally well-tolerated and associated with long progression-free survival, and have personally (senior author MG) observed that in the vast majority of these patients their CTCL recurs. In fact, only mogamulizumab has shown a (significant albeit small) effect on PFS.²⁶ With regard to the full potential of treating CTCL and other PTCLs, as well as any and all T cell malignancies that might have failed standard therapy and who are therefore considered for peripheral stem cell transplantation, we believe that our strategy may offer another therapeutic option in patients expressing V β 2.

26. Kim YH, et al. MAVORIC Investigators. Mogamulizumab versus vorinostat in previously treated cutaneous T-cell lymphoma (MAVORIC): an international, open-label, randomised, controlled phase 3 trial. *Lancet Oncol.* 2018 Sep;19(9):1192-1204.

Rev. 4, Point 4. The authors state that eliminating 50% of the T cell population with the Autolus approach would result in significant problems and immunodeficiency. What evidence that this would be an issue since the number of almost infinite TCR diversity would only be reduced by 50% which would likely not impact immune responses.

Please see response (and included references) addressing **Rev. 2, Point 1** (above) regarding comparison of our approach of CAR-T targeting the specific TCRV β family (which is estimated to eliminate 4-9% of the normal T cell population) to that of targeting TRBC1 and TRBC2 (which is estimated to eliminate up to ~50% of the normal T cell population). However, we agree that whether an ~10% T cell repertoire depletion (e.g. with our more specific anti-V β 2 approach) is better or equal to an ~50% T cell repertoire depletion remains an open question that requires long term follow-up and real world data that would take many years and include vaccination data and infection rate data (including shingles, covid, flu). Thus, we find it is noteworthy that the extent of CD4 T cell depletion in HIV-infected persons has been shown to correlate with the risk of a new AIDS event or death. A large study using data from >75,000 patients reported hazard ratios of 0.35 (0.30-0.40) for counts <200 cells/ μ l, 0.81 (0.71-0.92) for counts 200 to <350 cells/ μ l, 0.74 (0.66-0.83) for counts 350 to <500 cells/ μ l, and 0.96 (0.92-0.99) for counts \geq 500 cells/ μ l²⁷.

27. Young, J., et al. CD4 cell count and the risk of AIDS or death in HIV-Infected adults on combination antiretroviral therapy with a suppressed viral load: a longitudinal cohort study from COHERE. *PLoS Med.* 9(3):e1001194, (2012).

Rev. 4, Point 5. Killing curves shown in Figure 1h are very unimpressive. What is the reason for this? Could the CARTs be exhausted during manufacturing due to constant exposure of target (VB2+ T cells) during expansion?

The reviewer appears to be referring to Extended Data Fig. 1h. To be clear, this is an autologous CD8+ CAR-T cell killing assay presented to in fact demonstrate the suboptimal efficacy of autologous CAR-T cells for T malignancy treatment. This is in fact part of the

justification for our approaches (detailed throughout the manuscript) to overcome the T cell disfunction present in CTCL and other T cell malignancies. Furthering this point, we also repeated this autologous CAR-T killing assay with two additional patient donors, and consistently found limited killing efficacy of autologous CAR-T cells (shown below and in updated Extended Data Fig. 1h).

Rev. 4, Point 6. A major tenant of this paper is the need to generate “off-the shelf” TRAC KO CART since apparently endogenous (autologous T cells) from CTCL patients do not perform as well as normal allogeneic T cells or TRAC KO allogeneic T cells (results of Fig 1c/e vs Fig 1d). Is this conclusion made from a single patient and a single manufacturing generation? Is this the case for other PTCL patients? The only setting where this has been suggested is in CLL where even there the data is not definitive and only suggestive in spite of many samples and donors being analyzed. The need to move to off the shelf is justified by this but this is not demonstrated clearly since only a single patient is studied. The authors could do additional well controlled studies trying to confirm their preliminary observation and also no mechanistic studies are done to determine why this might be. Confirmatory experiments demonstrating the defect in autologous CTCL or PTCL CART compared to healthy or to allogeneic triple KO CART could also be tested with other CARs such as CAR19 and targets CD19+ B cell lines (Ramos etc) vs VB2 and VB2+ targets.

We appreciate the comments/references provided by the reviewer on T cell functional compromise in CLL²⁸. That the normal T cells are functionally compromised in CTCL is well documented in clinical data (increased risk of herpes zoster, Staphylococcus aureus, etc.¹²⁻¹⁷). That the normal T cells are compromised by number and T cell receptor repertoire²⁹⁻³¹ is also well documented as a fundamental feature of this malignancy. The compromised cell-mediated immune functionality of normal T cells in CTCL patients is due to a diverse set of immune activities imparted by the malignant T cells on the normal T cells, as summarized in Rook AH, et al. J Immunol (1995) 154:1491-1498 (see ref. 25 above), and further elucidated above in response to **Rev. 2, Point 8**.

Nonetheless, to provide additional relevant data in this regard, we repeated the autologous CAR-Vβ2 T cell killing assay using two additional patient donors, which also consistently showed compromised killing efficacy (below). These results are combined with our originally presented data and shown in a new Extended Data Fig. 1h (please see response to **Rev. 2, Point 4**).

28. Todorovic, Z, et al. CAR T Cell Therapy for Chronic Lymphocytic Leukemia: Successes and Shortcomings. *Curr Oncol.* 2022 May; 29(5): 3647–3657.

29. Yawalkar, N, et al. Profound loss of T-cell receptor repertoire complexity in cutaneous T-cell lymphoma. *Comparative Study Blood.* 2003 Dec 1;102(12):4059-66.

30. Yamanaka, K, et, al. Decreased T-cell receptor excision circles in cutaneous T-cell lymphoma. Clin Cancer Res. 2005 Aug 15;11(16):5748-55
31. Gleason, L, et, al. Reduced Overall T-Cell Receptor Diversity As an Indicator of Aggressive Cutaneous T-Cell Lymphoma. Blood (2022) 140 (Supplement 1): 3539–3540.

Rev. 4, Point 7. Triple KO experiments and humanization studies done in Figures 2 and 3 are well organized and the data is compelling. The issue is related to the lack of GvHD suggested by the authors. In Figure 4 there is no observation of weight loss of formal mouse GvHD staging. One would expect that the NC and lentiviral CARTVB2 to have significant weight loss and death since both have intact TCR. It is clear that the lentiviral CARTVB2 effectively clear tumor like the AAV KI and TRAC KO model but both lentiviral and AAV groups have similar anti-tumor effects (expected) but also the same survival (unexpected). So why is this and what are the GvHD scores for these groups of NSG mice? If lentiviral CARTVB2 do not induce any xenogeneic GvHD then this would be very interesting. Also the followup for these experiments is very limited. What is the survivals in the lentiviral and AAV groups after day +37. Longer follow-up would be essential to look at relapse rates (if relapsed tumors are antigen + or -) and GvHD effects.

To clarify for the reviewer, the presented in vivo data NC (No-Treatment Control) group did not receive a CAR-T cell transfer. For CAR-CD19 and lenti-hCAR-V β 2 groups, CAR-T cells underwent triple-KO and CD3+ cell depletion before being transferred into NSG mice. Therefore, neither lenti-hCAR-V β 2 T cells or AAV-hCAR-V β 2 T cells would be expected to induce GVHD effects in NSG mice. To the reviewer's point, we agree that longer follow-up may provide additional information regarding relapse rates as a key parameter for further comprehensive evaluation in preclinical models, but consider this beyond the scope of the current manuscript.

Rev. 4, Point 8. The data shown in Extended Figure 2 demonstrate high efficiency of TRAC deletion (>90%), and B2/Class I deletion (>95%) but low efficiency of Class II deletion (CIITA). Would be important using NHEJ sequencing what the actual deletion of CIITA is...may take some time for Class II to decrease at the protein level or simply that the editing is poor. Also, no off target analyses are presented in these cells subjected to multiplex editing with both HR and NHEJ approaches.

Regarding our selection and utilization of each engineering component of our triple-KO strategy, please see the detailed response above addressed in **Rev. 2, Point 5**.

Regarding MHC-II depletion, since CIITA acts as a transcription factor that promotes MHC-II expression, thus its KO may not completely eliminate MHC-II transcription. Nonetheless, this strategy has been shown to substantially reduce its expression by others (as referenced in the manuscript as ref. 29, Kagoya et al., 2020), and as measured by us (Extended Data Fig. 2).

Rev. 4, Point 9. The cytokine data presented in part in Figure 5 is difficult to interpret. Although the exercise is admirable there are no strong conclusions that can be made unless this were done many times with many different donors then the general conclusions summarized or proposed in Figure 5J. Also there is really limited meaning in general without any in vivo validation studies. For instance, one or two validation runs testing the best and worse cytokine combinations on in vivo killing would be really provide important context for all of these comprehensive in vitro studies.

Please see the detailed responses regarding the cytokine data addressed in **Rev. 1, Point 2** and **Rev. 4, Point 2**. We have now provided additional data using a limited set of cytokine combinations and cells from multiple donors (updated panels in **Fig. 5b** and **Fig. 5f**).

Rev. 4, Point 10. The data presented in Figure 6 is incremental at best and not well related to the rest of the focus of this manuscript and provides a distraction and not any unifying hypothesis or important mechanistic studies. Seems like a completely different topic.

Please see the detailed response (and provided references) addressed in **Rev. 2, Point 8** regarding the structure of this manuscript and our use of dual strategies (CAR-T and therapeutic antibody) to more specifically target T cell malignancies. We defer to the editor regarding final decisions on placement of figures.

REVIEWERS' COMMENTS

Reviewer #1 (Remarks to the Author):

The authors have adequately responded to reviewer comments

Reviewer #3 (Remarks to the Author):

This reviewer satisfied authors 'response and additional supplemental data in this resubmission.

Reviewer #4 (Remarks to the Author):

The revised manuscript was reviewed and the authors response to the critique was likewise reviewed in detail. The authors responded extensively to all of the reviewers criticisms and suggestions.

Major: There are many issues that are being highlighted in this manuscript. These many issues were raised by all reviewers. The focus on generating triple gene edited UCART to overcome the underlying decreased function of CTCL patients normal lymphocytes is a very important issue but not mechanistically pursued. The need for humanized CART is not well justified. The extensive studies identifying the optimal cytokine expansion for CART is not convincing and not well explored. Finally no comparative studies were performed comparing LV expression vs AAV KI for safety or expression or expansion or antitumor efficacy was performed. Also how this may be a better approach than TCRB1 or TCRB2 CART as per autolus is different, better or worse is not sufficiently expanded upon

Reviewer #4 (Remarks to the Author):

The revised manuscript was reviewed and the authors response to the critique was likewise reviewed in detail. The authors responded extensively to all of the reviewers criticisms and suggestions.

(1) The focus on generating triple gene edited UCART to overcome the underlying decreased function of CTCL patients normal lymphocytes is a very important issue but not mechanistically pursued.

This point was thoroughly addressed in the response to reviewers (Rev. 2, Point 4; Rev. 2, Point 8; Rev. 4, Point 6), including extensive references regarding mechanistic evidence of compromised cell-mediated immunity in CTCL patients.

(2) The need for humanized CART is not well justified.

We chose to humanize the antigenic receptor for our anti-V β 2 CAR-T as part of an overall strategy that involved the parallel generation of anti-V β 2 humanized antibody. In both cases, humanization is well-known to decrease immunogenicity upon transfer to patients. To explicitly highlight this point, we have added the following sentence and two references:

“CAR-T receptor humanization is a strategy shown to increase CAR-T survival after infusion.”

Temple WC, et al. Framework humanization optimizes potency of anti-CD72 nanobody CAR-T cells for B-cell malignancies. *J Immunother Cancer*. 2023 Nov 24;11(11):e006985. doi: 10.1136/jitc-2023-006985. PMID: 38007238

Song F, et al. Safety and efficacy of autologous and allogeneic humanized CD19-targeted CAR-T cell therapy for patients with relapsed/refractory B-ALL. *J Immunother Cancer*. 2023 Feb;11(2):e005701. doi: 10.1136/jitc-2022-005701. PMID: 36808074

(3) The extensive studies identifying the optimal cytokine expansion for CART is not convincing and not well explored.

This point was addressed in the response to reviewers (Rev. 1, Point 2; Rev. 2, Point 9; Rev. 4, Point 2; Rev. 4, Point 9), as well as with additional data and commentary in the manuscript. We believe that these datasets provide readers with insight into optimization of cytokine expression for CAR-T expansion following our generation protocol. To temper our conclusions in response to the reviewer’s comment, we have removed the following sentence from the Results section:

Based on hCAR-V β 2 expansion levels, persistent killing and re-expansion ability upon repeated CAR engagement (summarized in Fig. 5g), ex vivo culture conditions

containing IL7+IL12+IL15 should be considered for allogeneic hCAR-V β 2 T cell production.

(4) No comparative studies were performed comparing LV expression vs AAV KI for safety or expression or expansion or antitumor efficacy was performed.

This point was addressed in the response to reviewers (Rev. 2, Point 7). Please note that the possibility of LV integration induced secondary leukemia has been further raised as a potential risk with the recent FDA announcement of 19 reports of new blood cancers in anti-BCMA or anti-CD19 CAR-T treated patients.

(<https://www.fda.gov/vaccines-blood-biologics/safety-availability-biologics/fda-investigating-serious-risk-t-cell-malignancy-following-bcma-directed-or-cd19-directed-autologous>)

(5) How this may be a better approach than TCRB1 or TCRB2 CART as per autolus is different, better or worse is not sufficiently expanded upon.

This point was thoroughly addressed in the response to reviewers (Rev. 2, Point 1; Rev. 4, Point 1; Rev. 4, Point 4). We have already expanded upon this point in additional commentary in the Discussion:

A promising CAR-T approach to T cell malignancy is the targeting of one of the two potential T cell receptor beta constant regions, TRBC1 or TRBC2. Maciocia et al. have shown that the proportion of TRBC1+ T cells varies between 25 and 47% in healthy donors, regardless of the T cell subset⁶⁰. T cell leukemias and lymphomas, instead, are either clonally TRBC1 positive or negative. Therefore, TRBC1 CAR-T cells may specifically eliminate TRBC1+ malignancies and normal T cells while sparing TRBC2+ normal T-cells. A clinical trial testing TRBC1 CAR-T cells in T cell lymphomas is ongoing (AUTO4). Even so, this approach will result in substantial TRBC1+ normal T cell depletion and it is as yet unclear whether the residual T cell repertoire will be sufficient to maintain defense against pathogens and/or cancer cells. To overcome these potential drawbacks and circumvent potential risks, we sought to target the single specific TCR-V β expressed on each T malignancy.